

# A new method to quantify mineral dust and other aerosol species from aircraft platforms using single particle mass spectrometry

Karl D. Froyd[1,2], Daniel M. Murphy[1], Charles A. Brock[1], Pedro Campuzano-Jost[2,3], Jack E. Dibb[4], Jose-Luis Jimenez[2,3], Agnieszka Kupc[1,5], Ann M. Middlebrook[1], Gregory P. Schill[1,2], Kenneth L. Thornhill[6], Christina J. Williamson[1,2], James C. Wilson[7], Luke D. Ziemba[8]

[1] NOAA Earth System Research Laboratory Chemical Sciences Division Boulder, CO, 80305, USA
[2] Cooperative Institute for Research in Environmental Sciences, University of Colorado, Boulder, CO, 80309, USA
[3] Department of Chemistry, University of Colorado, Boulder, CO, 80309, USA
[4] Earth Systems Research Center, Institute for the Study of Earth, Oceans, and Space, University of New Hampshire, Durham, NH, 03824, USA
[5] Faculty of Physics, University of Vienna, 1090 Vienna, Austria
[6] NASA Langley Research Center, Science Systems and Applications, Inc., Hampton, VA, 23666, USA
[7] Department of Mechanical and Materials Engineering, University of Denver, Denver, CO, 80210, USA
[8] NASA Langley Research Center, Hampton, VA, 23681, USA

*Correspondence to*: Karl D. Froyd (Karl.Froyd@noaa.gov)

**Abstract.** Single-particle mass spectrometer (SPMS) instruments characterize the composition of individual aerosol particles in real time. Their fundamental ability to differentiate the externally mixed particle types that constitute the atmospheric aerosol population enables a unique perspective into sources and transformation. However, quantitative measurements by SPMS systems are inherently problematic. We introduce a new technique that combines collocated measurements of aerosol composition by SPMS and size-resolved absolute particle concentrations on aircraft platforms. Quantitative number, surface area, volume, and mass concentrations are derived for climate-relevant particle types such as mineral dust, sea salt, and biomass burning smoke. Additionally, relative ion signals are calibrated to derive mass concentrations of internally mixed sulfate and organic material that are distributed across multiple particle types.

The NOAA Particle Analysis by Laser Mass Spectrometry (PALMS) instrument measures size-resolved aerosol chemical composition from aircraft. We describe the identification and quantification of nine major atmospheric particle classes, including sulfate/organic/nitrate mixtures, biomass burning, elemental carbon, sea salt, mineral dust, meteoric material, alkali salts, heavy fuel oil combustion, and a remainder class. Classes can be sub-divided as necessary based on chemical heterogeneity, accumulated secondary material during aging, or other atmospheric processing. Concentrations are derived for sizes that encompass the accumulation and coarse size modes. A statistical error analysis indicates that particle class concentrations can be determined within a few minutes for abundances above ~10 ng m⁻³. Rare particle types require longer sampling times.

We explore the instrumentation requirements and the limitations of the method for airborne measurements. Reducing the size resolution of the particle data increases time resolution with only a modest increase in uncertainty. The principal limiting factor to fast time response concentration measurements is statistically relevant sampling across the size range of interest, in particular, sizes D<0.2 μm for accumulation mode studies and D>2 μm for coarse mode analysis. We demonstrate the use of a virtual impactor to enhance sampling statistics for the inherently sparse coarse mode. Performance is compared to other airborne and ground-based composition measurements, and examples of atmospheric mineral dust concentrations are given. The wealth of information afforded by composition-resolved size distributions for all major aerosol types represents a new and powerful tool to characterize atmospheric aerosol properties in a quantitative fashion.



## 1 Introduction

Particle mass spectrometry is a valuable method for characterizing atmospheric aerosol composition from airborne platforms. Instrumental techniques can be broadly categorized into bulk methods, where all aerosol within a size range are collected and characterized as a population (Canagaratna et al., 2007; Pratt and Prather, 2012), and single-particle methods that characterize

individual particles as a subset of the aerosol population, with a few hybrid methods also demonstrated (Cross et al., 2009; Freutel et al., 2013). Single-particle mass spectrometer instruments (SPMS; Hinz and Spengler, 2007; Murphy, 2007) have been used for over 25 years to characterize the chemical composition of atmospheric aerosol from ground sites and aircraft platforms. The NOAA Particle Analysis by Laser Mass Spectrometry (PALMS) instrument first flew in 1998 (Thomson et al., 2000), and several other SPMS instruments have successfully flown on airborne platforms (Brands et al., 2011; Coggiola et al., 2000; Pratt et al.,

2009; Trimborn et al., 2000; Zelenyuk et al., 2015). Their high sensitivity to a wide variety of aerosol species, size-resolved capability, and ability to characterize internally and externally mixed aerosol species make SPMS instruments well suited to airborne studies of atmospheric aerosol composition.

PALMS measures aerosol composition by evaporating and ionizing individual particles using a single pulse from a powerful laser, then analyzes the ions with a time-of-flight mass spectrometer. PALMS and other SPMS instruments that use single-step laser

desorption-ionization (LDI) are not inherently quantitative because ion formation is not a well-controlled process and gives rise to considerable particle-to-particle variability in both total and relative ion signals (Hinz and Spengler, 2007; Murphy, 2007). Many bulk aerosol mass spectrometers (Canagaratna et al., 2007; Tobias et al., 2000) and some SPMS instruments (Passig et al., 2017; Simpson et al., 2009; Sykes et al., 2002) use a two-step particle desorption and ionization process that can more readily quantify particle sub-components. Very high laser irradiances generate plasmas that can also improve consistency in ion signals but at the

expense of losing all molecular information (Wang and Johnston, 2006).

SPMS instruments have not typically calibrated the absolute ion signal intensity to aerosol mass abundance due to ionization variability. Also, SPMS particle detection usually relies on optical scattering, so that the overall detection efficiency is a strong and variable function of particle size. Relative abundance measurements of internally mixed aerosol sub-components have been reported in PALMS and other SPMS systems for metals (Cziczo et al., 2001; Murphy et al., 2007; Zawadowicz et al., 2015),

organosulfate species (Froyd et al., 2010; Liao et al., 2015), elemental carbon (Healy et al., 2012), and sulfate and organic material (Jeong et al., 2011; Middlebrook et al., 1998; Murphy et al., 2006; Zelenyuk et al., 2008; Zhou et al., 2016). Some groups have scaled SPMS data rates to aerosol reference instruments to derive total number or mass concentrations (Bein et al., 2006; Pratt et al., 2009; Qin et al., 2006) or concentrations for specific particle types (Gemayel et al., 2017; Healy et al., 2012; Jeong et al., 2011; Reinard et al., 2007; Shen et al., 2018). Many of these scaling studies invoke potentially large assumptions such as constant SPMS

detection efficiencies or a single density applied to all particles that can strongly affect derived concentrations. Uncertainties in such derived concentrations have not previously been reported. To date these methods have been restricted to ground-based sampling under relatively high aerosol loadings (~1-100 µg m$^{-3}$) and have employed long sample times ≥1 hr. Few coarse mode concentrations have been reported (Qin et al., 2006), and in particular, the authors are not aware of any studies using SPMS to determine absolute concentrations of mineral dust.

Mineral dust is one of the most abundant aerosol types in the atmosphere. Dust contributes a substantial fraction to global aerosol optical depth by scattering and absorbing radiation. Dust's role as a leading cirrus cloud nucleating agent (Cziczo et al., 2013) further elevates its importance for the climate system. However, measurement techniques for airborne studies, particularly fast-response online methods, are lacking. Online bulk mass spectrometry techniques are typically not sensitive to refractory particles such as dust. Also, instruments and aircraft inlets must be optimized to sample coarse mode aerosol up to several microns in size.

Electron microscopy (EM) techniques with associated elemental analysis remain valuable offline single-particle methods to detect



and quantify components associated with mineral dust (Kandler et al., 2009; Levin et al., 2005; Lieke et al., 2011; Matsuki et al., 2010) and other less volatile aerosol such as sulfate, sea salt, industrial metals, and some carbonaceous particles (Pósfai et al., 2003; Sheridan et al., 1994). Computer-controlled EM analysis can now characterize thousands of particles and generate population statistics of size, morphology, and detailed chemical composition (Ault and Axson, 2017; Craig et al., 2017). However,

continuous measurements at high time resolution remain impractical, the derivation of mass concentrations under background aerosol levels is challenging, and like most offline methods, volatile or reactive aerosol species can change prior to analysis.

A measurement gap remains for fast-response detection of mineral dust and other refractory or coarse mode particles. Additionally, size-resolved measurements and characterization of particle mixing state, i.e., the distribution of chemical constituents within single particles or across different particle types, are tractable by few methods. SPMS instruments are uniquely capable of detecting

both refractory and non-refractory particles in real time. PALMS and other SPMS instruments with sufficient laser power observe a chemical fingerprint for every type of aerosol particle in the atmosphere, minimizing chemical bias. Lower laser power and/or a longer ionization wavelength can result in biases against particles such as sulfate (Wenzel et al., 2003). SPMS instruments are particularly adept at characterizing some climate-relevant aerosol types, including mineral dust, biomass burning smoke, sea salt, and biological particles with high sensitivity and selectivity.

We present a new method that combines PALMS composition with independently measured particle size distributions to determine absolute number, surface area, volume, and mass concentrations of mineral dust, biomass burning, sea salt, and other common atmospheric particle types, with fast time response applicable to aircraft sampling where total mass concentrations are often >100 times lower than at ground level. Low detection limits on the order of 10 ng m$^{-3}$ for principal particle types are typical over a few minutes of sampling time. A unique capability of this technique is the derivation of number concentration for specific particle

types, which is particularly important for aerosol-cloud interaction studies. Size-resolved aerosol composition is measured over a wide size range that spans the accumulation and coarse modes under most atmospheric conditions. Additionally, we determine bulk-like mass concentrations for sulfate and organic material that are distributed across multiple particle types. The quantification methods described here are developed specifically for the PALMS instrument, but they are designed to act as a framework for quantifying particle types using other well-characterized SPMS instruments. We summarize the principal sampling considerations

and measurement criteria for deriving particle type concentrations, and we conclude with general recommendations for implementing the method in airborne composition studies. Estimations for principal sources of uncertainty are detailed in the Appendix.

## 2 Measurement Methods

### 2.1 Airborne aerosol sampling

Aerosol properties were measured aboard the NASA DC-8 aircraft during three campaigns: DC3, SEAC4RS, and ATom. The Deep Convective Clouds and Chemistry (DC3) campaign was based in Salina, KS in Apr-May 2012 and targeted convective outflow from isolated storm systems (Barth et al., 2015). The Studies of Emissions and Atmospheric Composition, Clouds and Climate Coupling by Regional Surveys (SEAC4RS) campaign was based in Houston, TX in Aug-Sept 2013, and sampled a variety of continental environments including regions with high biogenic activity, urban emissions, wildfires, and convective outflow

(Toon et al., 2016). The NASA Atmospheric Tomography (ATom) campaign consisted of four seasonal deployments from 2016-2018 to map the troposphere from near pole-to-pole in north-south transects along the Pacific and Atlantic basins (Wofsy et al., 2018). Measurements during the New England Air Quality Study (NEAQS) campaign were taken aboard the NOAA WP-3D aircraft based in Portsmouth, NH in July-Aug 2004, with flights targeting anthropogenic emissions from the eastern US (Fehsenfeld





et al., 2006). During the NASA Mid-latitude Airborne Cirrus Properties Experiment (MACPEX) campaign the WB-57 aircraft was based in Houston, TX in Mar-Apr 2011 and sampled tropospheric continental and stratospheric background air near cirrus cloud systems (https://espo.nasa.gov/macpex/). Cloudy flight segments are excluded from all aerosol data (Murphy et al., 2004b).

The airborne sampling methodology for ATom DC-8 deployments is detailed in Brock et al. (2019). Instruments are described in Sect. 2.2. For all DC-8 deployments most instruments used the University of Hawaii aircraft inlet operated at isokinetic conditions. This inlet was previously characterized to transmit aerosol particles ≥5.0 µm aerodynamic diameter with 50% efficiency at low altitude, and ≥3.2 µm at 12 km altitude (McNaughton et al., 2007). PALMS and particle size spectrometers subsample a minor flow from the main inlet flow. The PALMS instrument flow was 0.75 lpm, and particle spectrometer flows were 0.05-0.1 lpm. Particle spectrometers flows were actively dried using Nafion driers (Perma Pure), typically to <40% relative

humidity. Residence times between the aircraft inlet and instrumentation were 0.5-3.5 s. The AMS instrument used a dedicated HIMIL aircraft inlet (Stith et al., 2009) with residence times typically <0.5 s and no active drying. The SAGA filter samplers used the University of New Hampshire aircraft inlet that has similar particle transmission characteristics to the University of Hawaii inlet (McNaughton et al., 2007). Aboard the NOAA WP-3D during NEAQS aerosol particles were sampled using a low-turbulence inlet operated isokinetically (Wilson et al., 2004) and transmitted to size spectrometers inside the cabin. In the WB-57 during

MACPEX the FCAS instrument sampled particles using an anisokinetic inlet (Jonsson et al., 1995). PALMS was located inside a wing pod for NEAQS and inside the WB-57 nose for MACPEX, and for both campaigns PALMS sampled aerosol using a forward-facing anisokinetic tube that enhanced large particle concentrations.

    A virtual impactor was added upstream of PALMS for the DC-8 ATom deployments to enhance supermicron particle concentration. The design is based on Loo and Cork (1988) and is scaled to achieve an enhancement of at least 50% of the flow

ratio above ~2.0 µm. The virtual impactor was operated at total-to-minor flow ratios of 5.6 - 11. Figure S1 shows the virtual impactor design and its concentration enhancement for lab-generated aerosol.

## 2.2 Aerosol size and composition measurements

The NOAA Particle Analysis by Laser Mass Spectrometry (PALMS) instrument (Thomson et al., 2000) characterizes the size and chemical composition of individual aerosol particles from about 0.15-5 µm in diameter. Particles pass through an aerosol focusing

lens (Schreiner et al., 2002) and enter a vacuum where they pass through two continuous laser beams and scatter light. The transit time between the beams provides the particle velocity, which is used to determine particle aerodynamic diameters based on laboratory calibrations using polystyrene latex sphere size standards (Duke Scientific). A scatter signal triggers a 193 nm pulse from an excimer laser that ablates and ionizes a single particle. The very close proximity of detection and ionization beams (center separation ~100 µm) enables PALMS to obtain positive ion mass spectra for >90% of particle triggers. This high targeting

efficiency in PALMS minimizes chemical biases from particles of different shape or density that could have diverse trajectories. In general, particle mass spectrometers could have a low bias for non-spherical particles due to diverging particle trajectories (Huffman et al., 2005). Either positive or negative ions are analyzed with a time-of-flight mass spectrometer, with the polarity switched every few minutes during flight. Single-particle mass spectra are post-processed to classify each particle into a compositional type and calculate the relative abundance of particle sub-components.

Several optical particle spectrometer instruments were used for size distribution measurements. During NASA DC-8 sampling campaigns, the Laser Aerosol Spectrometer (LAS 3340, TSI, Inc) measured concentrations for particles from 0.1 to about 5 µm, above which the aircraft inlet transmission truncated the size distribution. Also on the DC-8, an Ultra-High Sensitivity Aerosol Spectrometer (Droplet Measurement Technologies) measured particles from 0.06 to either 0.5 or 1.0 µm, above which the LAS data were used. On the NOAA P-3 aircraft, the combination of a Lasair model 1001 (Particle Measurement Systems) and a custom-



built white light optical particle counter (WLOPC) measured particle concentrations from 0.12 to about 8 µm. On the NASA WB-57 aircraft, a custom built Focused Cavity Aerosol Spectrometer (FCAS II) measured particles from about 0.07-1.5 µm dry diameter (Jonsson et al., 1995; Wilson et al., 2008). All sizes are reported as ammonium sulfate diameters, and all concentrations are reported at standard conditions (1013 hPa and 273.15 K).

During DC-8 sampling campaigns a high-resolution time-of-flight aerosol mass spectrometer (HR-ToF-AMS; Aerodyne Research; Canagaratna et al., 2007; Nault et al., 2018) measured non-refractory, bulk aerosol mass composition at 1 Hz resolution with 100% transmission for vacuum aerodynamic diameter $100<D_{va}<500$ nm (50 and 770 nm at 50% efficiency; see DeCarlo et al., 2004 for the definition of $D_{va}$). Raw mass spectra were analyzed at 1 min intervals, yielding detection limits for organic and sulfate aerosol mass concentrations of 75 ng m$^{-3}$ and 10 ng m$^{-3}$, respectively, on average in the free troposphere. Also during DC-8 campaigns,

soluble ions were measured using the Soluble Acidic and Gases and Aerosols (SAGA) offline ion chromatography from aerosol filters (Dibb et al., 1999). Typical sampling times were 5-15 min with detection limits of ~10 ng m$^{-3}$. Filter data are excluded when cloudy periods exceed 20% of the sample time or over altitude ranges exceeding 3 km.

## 3. Deriving absolute concentrations

### 3.1 Composition-resolved size distributions from SPMS combined with particle size spectrometers

The general method of deriving quantitative abundance from single-particle composition data is introduced here and outlined in Fig. 1, with details described in the following sections. The approach combines size-resolved, single-particle composition from the PALMS instrument with a concurrent measurement of size-resolved absolute number concentration. Typically, an optical particle spectrometer (OPS) is used to measure the aerosol size distribution across the accumulation and coarse modes, although other sizing techniques based on electric mobility or aerodynamic diameter, or a combination of techniques, can be employed.

Mass spectra of individual particles acquired with PALMS are classified into one of several compositional categories. Aerodynamic diameters, $D_a$, for each particle are converted to volume-equivalent (geometric) diameters, $D_{ve}$, using particle densities and dynamic shape factors to match the OPS data. The particle classes are binned into size ranges that align with the particle size spectrometer. Then the fraction of each particle class within each size bin is multiplied by the average concentration within that bin. The resulting composition-resolved size distribution is integrated to give absolute number, surface area, or volume,

concentrations for each particle class. Mass concentrations for each particle type are determined by applying particle densities to the volume concentrations. Total sulfate and organic mass concentrations were derived from the non-refractory particle types.

Figure 2 shows two composition-resolved volume distributions measured from aircraft that are representative of two diverse atmospheric environments. The left panels give raw spectra counts as a function of size for each PALMS composition class, as well as the OPS volume distribution. The PALMS size bins are then aligned to the OPS, and fractional abundances in each size

bin are applied to the OPS volume distribution to generate the right panels.

The composition-resolved size distributions in Fig. 2b and 2d contain a wealth of information and represent a powerful set of tools to investigate atmospheric aerosol properties. The atmosphere consists of an external mixture of particle types, for which compositional size modes are clearly revealed. Three broad aerosol regions are apparent in the volume distributions: the accumulation mode at $D_{ve}<0.5$ µm consisting of mostly non-refractory particle types, a coarse mode at $D_{ve}>1$ µm dominated by

mineral dust and sea salt, and the inter-mode minimum at $0.5<D_{ve}<1$ µm that is a mixture of accumulation and coarse mode composition. Most particle classes extend to sizes beyond their principal mode. For instance, sea salt and mineral dust can also contribute significantly to submicron aerosol volume. In many environments, an extension of the accumulation mode continues to >1 µm where non-refractory particle types contribute to supermicron volume (Fig. 2d).



The practical limitations of this method mostly originate from the need in airborne studies to derive statistically significant composition measurements across the atmospherically relevant sizes within a reasonable sampling time (~1-5 min). The Fig. 2 examples required tens of minutes of sampling to populate nearly the entire accumulation and coarse mode size ranges with particle mass spectra. Although PALMS size range encompasses most of the accumulation and coarse volume modes, Fig. 2a and 2c show

how the outer ranges of each mode are not efficiently characterized. The white areas in Fig. 2b and 2d represent aerosol volume that is not allocated to any particle class. In most cases the unallocated volume is a minor fraction of the total, and the composition can be extrapolated to fill the entire mode, assuming composition remains constant. The following sections describe simplifying assumptions that allow faster measurements of particle type concentration while maintaining reasonable uncertainties. Uncertainties and limits of detection for particle type concentrations are detailed in Appendix A.

**3.2 Particle composition classes**

Each PALMS particle mass spectrum is classified as one composition type, e.g., mineral dust, or sea salt, sulfate-organic-nitrate mixtures, according to dominant spectral signatures. PALMS particle classification has been described previously (Cziczo et al., 2001, 2004; Froyd et al., 2009; Hudson et al., 2004), and definitions for nine principal atmospheric particle types are updated here. Chemical signatures, sizes, and other properties are listed in Table 1. These particle classes are defined so as to broadly capture

the main chemical components or identify a distinct aerosol source. It is important to note that PALMS and other SPMS particle type definitions are flexible and can be tailored to a particular environment or objective. Only positive ion mass spectra are used to categorize particles into these classes. The classification method uses empirical criteria based on relative peak intensities, and a spectra clustering algorithm (Murphy et al., 2003) is then used to refine particle sorting. Figure S2 shows representative particle mass spectra for all classes in Table 1.

The most abundant classes under most tropospheric environments are the sulfate/organic/nitrate (SO) internal mixtures and biomass burning (BB) particles. Particles classified as SO can be composed of primary or secondary material from a wide variety of sources but contain no biomass burning or other clear chemical markers that denote a particular source. Biomass burning particles are identified by a distinct potassium signature, abundant organic signatures, and a lack of crustal, marine, or industrial metals, based on the method of Hudson et al. (2004). The crucial potassium signature is stable over weeks of aging, and due to

PALMS extreme sensitivity to alkali metals, the potassium signal is observed above organic background peaks even when potassium constitutes <<0.1% of particle mass (Cziczo et al., 2001). The identification of these primary biomass burning particles is both highly sensitive and selective and does not deteriorate with particle aging. Single-particle information is critical to differentiating biomass burning potassium from other potassium sources that can confound bulk measurements (Legrand et al., 2016; Sullivan et al., 2019). Although secondary aerosol material is by definition distributed across many composition classes,

the SO and BB classes contain the vast majority of sulfate, organic, ammonium, and nitrate aerosol mass.

A variety of minor but important particle types contributes to the external aerosol mixture of the lower atmosphere. The elemental carbon (EC) class include particles dominated by $C_n^+$ mass spectral signatures and are interpreted as mostly EC by mass (presumably black carbon) since small amounts of internally mixed organics will obscure the EC signatures. Particles with minor EC content are therefore not distinguishable by PALMS and are instead classified as SO. Some EC particles contain potassium,

which suggests a biomass burning source, and when accompanied by organic signatures these particles are instead classified as BB. Sea salt is easily distinguished by a dominant sodium signal, often with calcium, strontium, other alkalis, and sodium chloride ion clusters, but without crustal metals. Mineral dust (MD) spectra are identified from multiple crustal metal signatures such as silicon, aluminum, iron, and calcium and often contain trace amounts of alkalis, barium, tin, antimony, or lanthanides. This category is more heterogeneous than other classes and contains many different sub-types of spectra, representing a wide variety of



mineralogies. Meteoric material (MT) is identified by iron, nickel, and magnesium within particular intensity ratios (Cziczo et al., 2001) and without other crustal material, and is usually accompanied by strong sulfate signatures. The alkali salt (KS) category is reported here for the first time. The spectra for this class contain potassium and other alkali metals but without crustal material, and very low organic signatures distinguish them from biomass burning particles. Despite the similarity to biomass burning

particles they are not enhanced in smoke plumes. The spatial and vertical patterns of these particles suggest primary continental emissions, but their exact source is still uncertain. Their size is exclusively submicron, which suggests they are not a type of mineral dust. The KS class constitutes 0.1-0.5% of accumulation mode particles over the US and <0.1% in the remote troposphere. Heavy fuel oil combustion particles (FO) are readily identified by strong vanadium signatures mixed with sulfate, organics, and sometimes iron or nickel (Ault et al., 2010; Divita et al., 1996). Spectra not identified as any of the above composition types are

compiled into a class labeled as "Other" (OT), which contains a variety of minor particle types. By far the most abundant subtype in OT is a sulfate-organic mixture with possible alkali or metallic signatures that are small and difficult to distinguish from organic peaks. Other examples include spectra with the pyridinium ion and other amine signatures, industrial metals without obvious crustal components, and several types of organic-rich particles with distinct signatures that suggest unique but unknown sources. Primary biological particles are currently identified from negative ion spectra only (Zawadowicz et al., 2017), and a separate

particle class is not implemented. In continental air they account for ~1% of supermicron particles and <0.1% of all detected particles (Zawadowicz et al., 2019).

All particle types acquire secondary material such as sulfate, ammonium, organics, and nitrate during atmospheric transport and aging. This secondary accumulation does not change particle assignments, except that heavy coatings may partially obscure unique signatures, resulting in a particle classified as "Other". For example, a mineral dust particle that contains secondary sulfate, nitrate,

and organic material will still be classified as mineral dust, and the derived dust mass includes the secondary material. Similarly, BB particles may contain secondary material sourced from biomass burning and non-biomass burning emissions. Laboratory calibrations of secondary mass spectral signatures could be used to subtract secondary mass from primary particle types. In some cases, the chemical component that identifies a particle's source is a minor constituent. For example, particles in the meteoric class are mostly sulfuric acid by mass, and the metals from ablated meteorites only account for a few percent of mass. Similarly,

particles from heavy fuel oil combustion are composed of mostly sulfate and organic material but also contain the trace vanadium and other metals that denote their unique emission source.

### 3.3 Simplifying the size distribution

It is infeasible to retain the raw size resolution of the optical particle spectrometer (OPS) for the integrated concentration analysis, since some common commercial instruments report up to 50 size bins per decade of diameter. For example, to achieve a minimal

compositional representation with >5 particle spectra in each size bin would require >5000 spectra if acquired evenly across the instrument's size range. Accounting for inefficient acquisition at the extreme size limits of the instrument and with typical single particle mass spectra data rates of a few Hz, this would require sampling times >1 hour to display composition at the native resolution of the OPS. Therefore, raw size bins of the optical spectrometer must be combined into fewer bins to improve time resolution but at some expense in accuracy of the derived concentrations. Ideally, size bins are defined such that the composition

is homogeneous within each combined bin, in which case this simplification is rigorous and introduces no error to the derived number, surface area, and volume concentrations. Induced error should be minimized by defining size bins such that neither the concentration nor particle type fractions have strong gradients across a bin limit. Concentration products cannot be determined if zero PALMS spectra are acquired within any one size bin that contributes significantly to the integrated concentration from the



OPS. In practice, composition gradients across size bins and statistical noise at the size range extremes generates systematic error that increases as size bins are combined and the size distribution simplified.

To estimate the systematic errors associated with this approach, integrated volumes were calculated for a number of cases where composition was constant over an extended flight period. For each case PALMS particle class volumes were first determined at a high size resolution of ~20 bins per decade of diameter. Nearest neighbor diameter bins were then combined, and integrated particle volumes were recalculated for each particle class at the lower size resolution. When the total number of bins was reduced to three or four, the diameter limits were empirically defined based on volume modes and composition gradients, e.g., one or two bins across the accumulation mode from about 0.06-0.5 µm, one across the inter-mode minimum at 0.5-1.1 µm, and one coarse mode bin at 1.1-5 µm (vertical grey lines in Fig. 2b and d). The high resolution analysis is treated as a reference value, and the average deviation of derived volumes as a function of final bin count is plotted in Fig. 3a for diverse cases across several flight campaigns. Typical deviations are 5-25% when the size distribution is represented by three or four bins.

For sampling times of a few minutes, reducing the size resolution improves the data coverage, Fig. 3b. For each flight campaign, the number of sampling periods with sufficient statistics to generate concentration products are plotted relative to the 2-bin case. To generate concentration products, every diameter bin that contributes significantly to the total volume must include ≥5 PALMS spectra. The 3- or 4-bin approach offers a good tradeoff between reasonable time resolution and data coverage (65-85% for 3 bins) while still yielding particle volume and mass concentrations whose systematic errors are less than or equal to typical volume uncertainties of a particle spectrometer (Kupc et al., 2018). Small particle detection efficiency (see Sect. 3.6) was worse for DC3 than for SEAC4RS, resulting in a steeper reduction in data coverage between 3 and 4 diameter bins. For DC3 the lower sensitivity to small particles results in more sampling periods with <5 particles in the smallest size bin, so that relative data coverage with 4 diameter bins is 25-50% compared to SEAC4RS with 65-75%.

Fig. 3a suggests that induced errors are not a simple function of a particular atmospheric environment. Instead, most of the variability for any given bin count is due to sparse data within a single size bin or composition inhomogeneity across a size bin limit. The choice of size binning and time resolution when deriving integrated products can be altered based on both these conditions. Three size bins, where one bin encompasses the entire accumulation mode, is adequate for many tropospheric sampling environments (Fig. 2b). When certain external mixtures are apparent, it is recommended to split the accumulation mode into two size bins. For instance, biomass burning particles and meteoric or other stratospheric particles occupy the larger end of the accumulation mode, D>200 nm, whereas secondary sulfate/organic particles typical of the UT can be smaller, ~60-150 nm (Fig. 2d).

The PALMS size range encompasses the majority of accumulation and coarse mode size ranges under most atmospheric conditions (Fig. 2). Exceptions include the marine boundary layer and strong mineral dust plumes that often contain particles larger than ~4 µm, and very clean upper tropospheric conditions, where number and volume contributions can be significant for sizes below the PALMS range, e.g., $D_{ve}$ < 150 nm. In the latter case for example, the lower size bin limits for the concentration analysis can be set to fully include the lower end of the accumulation mode, e.g., $D_{ve}$(bin1) = 60-250 nm. The PALMS composition averages applied to that bin will be biased to the larger end of the bin ($D_{ve}$ ~150-250 nm), but PALMS fractions are applied to the entire bin as usual. The total concentration is still accurately measured by the particle spectrometer, but the PALMS composition is effectively extrapolated to sizes outside of the PALMS size range. The inherent assumption is that the composition across the lower half of the accumulation mode is homogeneous. While this extrapolation can be appropriate for many tropospheric and stratospheric environments, care should be taken in cases where the accumulation mode is weak and shifted to small diameters, such as very clean upper tropospheric conditions, or in heterogeneous environments, such as active aerosol emission sources mixing with background air.



### 3.4 Response of optical particle spectrometers (OPS) to composition

The optical scattering response of an aerosol particle depends on its size, shape, refractive index, and the light collection geometry of the spectrometer instrument. Size distributions derived from optical particle spectrometers are based on the assumption of a fixed refractive index and spherical shape to translate the optical response of the measured particle population into volume-equivalent diameter, $D_{ve} \equiv D_{opt}$. A typical atmospheric air mass contains an external mixture of several diverse particle types. Fortunately, the real refractive indices for background tropospheric environments have been observed to fall within a narrow range (n ≈ 1.50 - 1.56 at mid-visible wavelengths) such that ambient particle size measurements are not strongly affected by this assumption (Hand and Kreidenweis, 2002; Liu et al., 2008; Müller, 2002; Reed Espinosa et al., 2017; Shingler et al., 2016; Yamasoe et al., 1998). Ammonium sulfate and ammonium nitrate (n=1.53 and 1.56 at λ=532 nm, respectively) are common inorganic constituents. Visible refractive indices (n-ki) have been derived for mineral dust aerosol from a variety of field measurements and typically range from about n = 1.52-1.58 and k~0.001-0.01, (Balkanski et al., 2007; Dubovik et al., 2002; Kandler et al., 2011; McConnell et al., 2010; Müller et al., 2010; Petzold et al., 2009; Schladitz et al., 2009), with absorption increasing at shorter visible wavelengths. Pure mineral samples can exhibit higher variability. Many retrievals for ambient organic aerosol material fall within typical inorganic refractive indices (Aldhaif et al., 2018; Kassianov et al., 2014; Reed Espinosa et al., 2017; Shingler et al., 2016), although some laboratory surrogate species and a few atmospheric organics can have a wider range, n≈1.47-1.65 (Dinar et al., 2007; Dubovik et al., 2002; Hoffer et al., 2006; Rizzo et al., 2013; Schkolnik et al., 2007). Sulfuric acid, which is abundant in the stratosphere and sporadically in the troposphere, has a much lower refractive index of n=1.44 at 532 nm (Luo et al., 1996). Sulfuric acid aerosol also retain water (n=1.33 at 532 nm, Daimon and Masumura, 2007) even at low RH, making it a predominant outlier to typical refractive indices. Kupc et al. (2018) investigated the potential systematic error in prescribing the wrong refractive index to a representative UTLS aerosol population. The difference in aerosol volume assuming pure ammonium sulfate (n=1.54) versus sulfuric acid (n=1.44) was only 12%, which is lower than a typical aggregate volume uncertainty.

Few atmospheric particle types are strongly absorbing, and the complex index of refraction (k) for an ambient population is assumed to zero. Several exceptions follow. Particles containing elemental carbon (EC), presumably in the form of black carbon (BC), typically account for <1% of accumulation mode mass in the background atmosphere but up to ~10% inside wildfire plumes (Andreae and Merlet, 2001). The refractive index of pure black carbon has high variability, e.g., with n-ki = 1.74-0.44i (Hess et al., 1998), 1.95-0.79i (Bond and Bergstrom, 2006), and 2.26-1.26i (Moteki et al., 2010). Although their measured diameters can be erroneous by >10% (Kupc et al., 2018), the net error on total aerosol number and mass is typically much lower due to their small relative population. Hematite mineral also has a unique refractive index (2.5-1.0i at 405nm; Sokolik and Toon, 1999), but pure hematite particles are extremely rare in the atmosphere.

Ammonium sulfate and ammonium nitrate are convenient materials to calibrate optical particle spectrometers for use in atmospheric sampling. If generating monodisperse aerosol at coarse mode sizes is impractical, polystyrene latex sphere (PSL) size standards can be used, but their refractive index (n=1.59 at 532 nm; Ma et al., 2003) is not representative of typical atmospheric aerosol. Therefore, the scattering response of the sizing instrument to PSL particles must be converted to atmospherically relevant particles using Mie theory. Figure 4a shows Mie scattering intensities calculated for the LAS instrument at λ=663 nm for PSL and ammonium sulfate particles. At each diameter, the scatter intensity for a PSL particle is located on the ammonium sulfate intensity curve, and the associated ammonium sulfate diameter is determined. For a given scatter response the calibrated PSL diameter can shift up to 20% due to the different refractive index of ammonium sulfate. The inset in Fig. 4a shows an expanded region from D=0.8-2.4 μm, where oscillations render optical particle sizing more uncertain. These oscillations introduce additional sizing uncertainty that is inherent to all monochromatic particle spectrometers in the range where, depending on the collection geometry,



particle diameters are about 1-2 times the laser wavelength. Scattering intensity curves were smoothed so that each PSL diameter yielded one unique ammonium sulfate diameter. A raw PSL calibration curve is shown in Fig. 4b along with an ammonium sulfate-equivalent calibration curve derived using the diameter ratio curve. Also plotted in Fig. 4b are points from laboratory measurements of monodisperse ammonium sulfate particles generated with a differential mobility analyzer for D<1 μm. Closure

between the PSL-derived and measured ammonium sulfate response was not fully achieved. However, rapidly effluoresced ammonium sulfate particles are slightly non-spherical, and as a result the volume-equivalent diameter of the monodisperse ammonium sulfate aerosol is smaller than the mobility diameter, $D_{ve} < D_{mob}$. Agreement between measured and derived ammonium sulfate calibration curves improved after correcting the calibration diameters using shape factors of $\chi_t =1.03$ to 1.09, increasing with size (Huffman et al., 2005; Zelenyuk et al., 2006b). We confirmed that the ammonium sulfate particles had fully effloresced

in the calibration system by observing an abrupt increase in apparent optical size due to change in refractive index when the relative humidity was reduced below the efflorescence point.

Highly non-spherical particles such as some mineral dusts and black carbon aggregates have different scattering intensities and phase functions compared to their volume-equivalent spheres. The effect on the angular scattering pattern is not consistent across different shapes and cannot be described by any simple measure of asphericity (Curtis et al., 2008; Mishchenko et al., 1997; Moteki

et al., 2010; Peter and Michael, 1988). In general, the wide variety of atmospheric mineral dust morphologies will produce a diverse scattering response for particles with the same $D_{ve}$, with the principal effect of increasing the uncertainty in optically measured diameters. The degree to which sizing accuracy is affected depends on the degree and distribution of particle asphericity, surface roughness, the local steepness of the scattering intensity curve, and the angular collection geometry of the spectrometer. For particles that are freely rotating and not aligned with the instrument sample flow, as is the case near atmospheric pressure,

optical mis-sizing may be minimized because the scattering response is an average of multiple particle orientations. In the current treatment all particles are assumed to be spheres for the purposes of optical sizing.

### 3.5 Particle densities and dynamic shape factors

PALMS measures the aerodynamic diameter, $D_a$, for >90% of the chemically analyzed particles. For each particle $D_a$ is converted to a volume-equivalent diameter, $D_{ve}$, using the particle density ρ, dynamic shape factor χ, and the Cunningham slip correction

factor, $C_c$.

$$D_{ve} = \left(\frac{\chi \rho_0 C_c(D_a)}{\rho C_c(D_{ve})}\right)^{\beta} D_a \tag{1}$$

In the limit of continuum flow $\beta=0.5$, and for free molecular flow $\beta=1$ (DeCarlo et al., 2004). Particles exit the aerodynamic focusing lens at approximately 26 Torr and accelerate into a vacuum region at ~0.2 Torr where their aerodynamic size is measured by the time difference in scattering events from two laser beams spaced 33.1 mm apart. During acceleration, particles have

Knudsen numbers of 1-20 and therefore experience flow that is near the free molecular regime. A comprehensive model that considers transitional flow is used to convert aerodynamic diameter measured by PALMS to $D_{ve}$ (Murphy et al., 2004a). For submicron particles the model predicts $\beta>0.89$, such that the measured aerodynamic diameter is near the free molecular limit ($D_a \sim D_{va}$). As size increases beyond ~1 μm, particle motion in the PALMS inlet becomes more transitional, e.g., $\beta=0.75$ for 3 μm. The dynamic shape factor under these conditions for most particles is close to the vacuum shape factor $\chi_v$, which can deviate

significantly from shape factors measured at atmospheric pressure (Alexander et al., 2016; Dahneke, 1973a, 1973b). We further discuss free molecular shape factors for mineral dust below. Dry particle densities and shape factors for each composition class are listed in Table 1. Particle density for each measured particle is determined using one of three methods: 1) prescribed based on



literature values for the observed particle type, 2) calculated using prescribed values for pure particle sub-components and their relative component abundance, or 3) estimated here using simultaneous optical and aerodynamic measurements.

Optical scattering intensities are measured in PALMS as particles pass through two detection lasers. Scatter intensities are not directly used for individual particle sizing for several reasons: 1) the particle stream is wider than the laser cross-section so that

particles experience inconsistent laser intensities, 2) Mie oscillations produce a relatively flat scattering intensity from 0.5 – 1.0 μm, and 3) the photomultiplier signals begin to saturate at $D_{ve} > 0.6$ μm. However, when averaged over hundreds of particles, the scattering response yields information about particle density, shape, and refractive index for submicron sizes (Moffet and Prather, 2005; Murphy et al., 2004a). Figure 5 shows simultaneous PALMS measurements of optical scattering intensity ($I_{scat}$) and aerodynamic diameter from one scattering laser for several different particle classes. For a given $D_{ve}$, a higher particle density

translates into larger $D_a$ values, yielding an $I_{scat}$-$D_a$ curve further toward the right side of the graph. A larger shape factor will shift curves further to the left.

After converting each particle's $D_a$ to $D_{ve}$, the scattering curves converge for $D_{ve} < 0.5$ μm (Fig. 5b). A single relationship is expected between physical diameter and optical scattering intensity for all spherical particle types with similar refractive index. The $I_{scat}$-$D_a$ analysis in Fig. 5b helps validate prescribed densities and shape factors for known particles and also provides guidance

for unknown particle types. Particle types with known density and shape factors, such as pure sulfuric acid, ammonium sulfate, and known organic species act as internal standards.

All particles in the non-refractory classes (SO, BB, MT, FO) are internal mixtures of sulfate, organic material, and other minor components. Density is calculated for each particle in these classes as a weighted average of the pure component densities using the measured organic-to-sulfate mass fraction (see Sect. 3.7). Nitrate content is not considered in deriving density because nitrate

is difficult to differentiate from ammonium and other nitrogen species in PALMS positive ion spectra. In regions where ammonium nitrate is the dominant aerosol constituent, particle density and water content can be similarly calculated (Clegg et al., 1998) using an estimated aerosol nitrate or total nitrogen calibration.

Density values for ammonium sulfate-water and sulfuric acid-water solutions are calculated at the temperature and RH of the instrument inlet (Clegg et al., 1998; Vehkamäki et al., 2002; Wexler, 2002). Sulfuric acid can retain ~10-20% water even under

very dry (RH<<1%) sampling conditions, although additional water evaporation will take place in the PALMS low pressure inlet and vacuum region (Murphy, 2007; Zelenyuk et al., 2006a). The density for pure organic material is prescribed as 1.30 g cm⁻³ for SEAC4RS (Fig. 5) and other continental sampling campaigns, approximately the middle of a range of typical values observed in continental air, 1.2-1.45 g cm⁻³ (Cross et al., 2007; Turpin and Lim, 2001; Vaden et al., 2011; Zelenyuk et al., 2010, 2015). Organic density increases with oxidation level (Kuwata et al., 2012) as particles age in the atmosphere. Consequently, for the ATom remote

troposphere an $I_{scat}$-$D_a$ analysis like Fig. 5 indicates higher average organic densities of 1.35-1.45 g cm⁻³. The density of organic material in biomass burning has a similar range, 1.2-1.45 g cm⁻³, (Reid et al., 2004; Zelenyuk et al., 2015; Zhai et al., 2017) and 1.25 g cm⁻³ is prescribed here for continental US sampling and 1.35 g cm⁻³ for the ATom campaigns.

Refractive index differences affect the vertical position of the curves and shift the size where the optical response flattens out. Mie scattering intensity curves are plotted to demonstrate the effect of refractive index in Fig. 5c. For the PALMS wavelength and

collection geometry, increasing real and decreasing imaginary refractive index gives increased scatter intensity for D<0.5 μm. Large deviations beyond the typical range of 1.44-1.54 for atmospheric constituents are required to shift the response curve beyond typical variability. Elemental carbon stands out due to its large real refractive index and strong absorption. For most particle types, the $I_{scat}$-$D_a$ curves are far more sensitive to density and shape parameters than refractive index.

Elemental carbon (EC) particles are assumed to be composed of black carbon (BC), and particle density is prescribed at 1.8 g cm⁻

³ based on the density for pure BC of 1.8-2.1 g cm⁻³ (Bond and Bergstrom, 2006; Lide, 2016; Park et al., 2004). Shape factors for





EC particles in this size range will vary widely depending on the morphology of aggregates. Uncoated BC particles have $\chi_t$ values that range from 1.0 to >3.0 depending on size (Khalizov et al., 2012; Slowik et al., 2007). $\chi_v = 2.0$ is prescribed here, based on the assumption that relatively pure EC may exist as chain aggregates that have not fully collapsed into a quasi-spherical shape (Schnitzler et al., 2014). The density and shape factors for EC are less well constrained than other particle classes since the unique

refractive index renders the $I_{scat}$-$D_a$ analysis ineffective (Fig. 5b), and uncertainties in derived concentrations are accordingly higher.

Sea salt aerosol, when fully dehydrated to anhydrous inorganic salts, has a density of 2.1-2.2 g cm$^{-3}$ (Lewis and Schwartz, 2004; Zelenyuk et al., 2005). However, dried ambient sea salt particles have lower densities due to retention of water even after efflorescence (Cziczo et al., 1997; Shinozuka et al., 2004; Tang et al., 1997; Weis and Ewing, 1999) and to internally mixed organic

material. In the marine boundary layer where most sea salt is sampled, particles exist as solution droplets since the ambient RH is greater than the efflorescence RH (ERH) of 40-45% (Cziczo et al., 1997; Tang et al., 1997). Upon sampling, particles are dried to RH<<45% in the sampling lines, yet water does not fully evaporate. The $I_{scat}$-$D_a$ analysis yields a sea salt aerosol density/shape factor ratio of 1.45, which is close to the 1.41 g cm$^{-3}$ density of a spherical sea salt-water particle at the ERH of ~45% (Tang et al., 1997; Zhang et al., 2005). This suggests that efflorescence did not occur during the 0.5 s residence time between sampling and

analysis and that most sea salt in PALMS is analyzed as a metastable solution. Sea salt density is prescribed at 1.45 g cm$^{-3}$ with a shape factor of $\chi_v = 1$. Occasionally during research flights, particles were sent through a 300 °C thermal denuder for 3.3 s prior to analysis. The $I_{scat}$-$D_a$ analysis indicates that the heated sea salt fully effloresced and, assuming a dynamic shape factor of $\chi_v = 1.08$ (Beranek et al., 2012), had a density of 1.8 (see Fig. S3). We presume that efflorescence was complete during the 2-3 s residence time in the actively dried transfer tubing prior to sampling with optical particle spectrometers.

Mineral dust particle densities for the large majority of crustal minerals are typically 2.5-2.65 g cm$^{-3}$ (Davies, 1979; Kandler et al., 2009; Linke et al., 2006), with a few subtypes such as hematite having higher values. Shape factors are invariably $\chi_t>1$ with a typical range of 1.3-1.5 (Davies, 1979; Kulkarni et al., 2011; Linke et al., 2006). $\chi_t$ denotes the dynamic shape factor measured at atmospheric pressure, which is in transitional flow but near the continuum flow limit, i.e., $\chi_t(1\ atm)\sim\chi_c$. Most ambient mineral dust is coated by secondary organic and inorganic material, which reduces both particle density and shape factor. Preliminary

PALMS laboratory studies indicate that typical organic coatings add 5-10% to the mass of ambient dust particles, which reduces the density of a 2.65 g cm$^{-3}$ dust particle to 2.4-2.5 g cm$^{-3}$ and will also reduce the shape factor slightly. However, prescribing a density of 2.5 and shape factor of 1.4 yields $I_{scat}$-$D_a$ curves for dust that are clearly not consistent with other particle types (Fig. S3). To achieve internal consistency, dust must either be prescribed an implausibly low particle density of ~2.0 g cm$^{-3}$ or a larger shape factor. Based on the $I_{scat}$-$D_a$ analysis a density of 2.5 g cm$^{-3}$ and $\chi_v$ of 1.6-1.8 is prescribed to all ambient dust particles.

Large shape factors for ambient mineral dust are discussed in Appendix B.

Scattering intensities and sizes for additional particle types from Table 1 are shown in Fig. S3. Biomass burning particles are ~80-90% organic material by mass soon after emission, (Cubison et al., 2011; Levin et al., 2010; May et al., 2014) and they acquire sulfate, ammonium, and nitrate upon further aging, whereby density increases to 1.4-1.5 g cm$^{-3}$. In the stratosphere meteoric particles are nearly pure sulfuric acid (~1.7 g cm$^{-3}$) with small meteoritic inclusions, and they acquire organic material upon mixing

into the troposphere, whereby their density decreases. Particles from heavy fuel oil combustion are mostly composed of mixed sulfate and organic material with trace industrial metals and typically have a density of 1.3-1.6 g cm$^{-3}$. As stated above, for the concentration analysis, individual particle densities for these three classes (BB, MT, FO), as well as sulfate/organic/nitrate particles (SO) are calculated from relative sulfate and organic mass (see Sect. 3.7). Alkali salts have a density-shape factor ratio of approximately 1.5, which assuming spherical shape, is well below the 2.0-2.6 g cm$^{-3}$ density typical of crystalline alkali sulfates,




chlorides, carbonates, or oxides. It is possible that like sea salt, these alkali salt mixtures have not fully effloresced during sampling, and the retained water lowers their particle density.

**3.6 Detection efficiency**

The efficiency of acquiring single particle mass spectra from aircraft platforms depends on many factors including sampling biases
of the aircraft inlet, losses in sample tubing, transmission through critical orifices and focusing lenses, particle beam dispersion upon entering vacuum, sensitivity of the optical detection system, and targeting accuracy with the desorption/ionization laser. In practice the detection efficiency at small sizes ($D<\sim0.3$ µm) is limited by the optical scattering signal-to-noise and is a strong function of size, and large particle ($D>\sim1.5$ µm) detection is limited by impaction losses in tubing and instrument inlets. Detection efficiency is not used directly in deriving particle type concentrations presented here, yet it is an important diagnostic to assess
SPMS performance over the dominant size modes in the aerosol distribution.

PALMS detection efficiencies are calculated for flight segments with fairly constant and low concentration (about $<0.1$ µm$^3$ cm$^{-3}$), when the particle rate is not limited by hardware or software and therefore dead time is minimal. Measured $D_a$ is converted to $D_{ve}$ for all particles, and the total counts in each size bin are converted to an apparent concentration using the PALMS flow rate and sampling time. The PALMS observed concentrations are divided by OPS concentrations to determine detection efficiency.
Figure 6 shows detection efficiencies for two airborne campaigns. Although the curves show that average instrument performance is similar across different campaigns, variability within and between individual flights can be large. Within any given flight, the detection efficiency at a particular size routinely varies by x2-x5, and variations of >x10 are not uncommon between flights. This variability is due to a variety of the factors listed above, many of which change with ambient pressure. In particular, changes in the overlap between the particle beam and laser beams will dramatically affect the detection efficiency for all sizes or a range of
sizes. This variability in detection efficiency affects all particle classes nearly equally for a given particle size. A reduced detection efficiency does not directly impact particle type concentrations, but it can increase the statistical uncertainties and sampling time required to generate concentration products.

A tempting alternative to the method presented here (combining SPMS data with coincident size distribution measurements) is to determine the SPMS particle detection efficiency as a function of size under controlled conditions, and then multiply this curve by
the airborne size-dependent data rate to yield a quantitative particle concentration, similar to SPMS scaling methods used at ground sites (Bein et al., 2006; Jeong et al., 2011; Pratt et al., 2009; Qin et al., 2006; Shen et al., 2018). However, this approach is not recommended due to many possible pitfalls and large, unquantifiable errors. The key drawback is that the detection efficiency curve for PALMS and other SPMS instruments is extremely steep as it ascends several orders of magnitude across the accumulation mode from $D\sim0.1$ to $0.5$ µm, where particle number concentrations are also changing by orders of magnitude. The multiplication
of these two strong functions, combined with the inherent variability over different ambient conditions and instrument alignment, will produce large and intractable uncertainties, e.g., >x10 in mass. Other problems include the following:

- Both the measurement of detection efficiency and its application to derive concentrations are only valid when the particle data rate is not artificially limited by software or hardware or when instrument dead time can be accurately determined. In practice, aerosol concentrations in many lower tropospheric conditions and particularly inside plumes are high enough
35       that SPMS systems will far exceed their maximum acquisition rate and generate erroneously low concentrations.
- Small changes to the alignment of the particle beam with the detection lasers, ionization laser, and ion extraction optics, which are not uncommon on airborne platforms, have a large effect on particle detection efficiency. Detection efficiency must also be re-determined after any routine alignment adjustments. For PALMS the second detection laser beam is only $\sim150$ µm wide.



- The flow characteristics of pressure reduction orifices change with upstream (ambient) pressure, changing particle trajectories downstream of the orifice in ways that are very sensitive to physical alignment and may not vary smoothly with pressure. One example is that pressure-controlled inlets can act as virtual impactors that enhance concentrations above a certain particle size that is both difficult to define and changes with pressure.

- Upstream pressure reduction orifices routinely accumulate small amounts of aerosol material that subtly change particle trajectories with large impacts to detection efficiency (Fig. 6).

The approach described in Sect. 3.1, mapping the PALMS composition measurements to independently measured size distributions without the need to determine size-dependent detection efficiencies, circumvents these complications.

### 3.7 Sulfate and organic mass concentrations

10       In addition to deriving concentrations for individual particle classes, the subcomponents of internally mixed particles can also be quantified. Signal intensity ratios in PALMS mass spectra for components of interest are calibrated to known mass fractions in laboratory generated aerosol standards. By combining these calibrations with the particle class concentrations described here, absolute mass fractions for aerosol subcomponents such as sulfate and organic material can be determined. For instance, the average sulfate mass fraction is first determined for all non-refractory particle classes using mass spectral signal ratios, then this

mass fraction is multiplied by total mass concentration of those particle classes to yield an aerosol sulfate mass concentration. The resulting mass concentrations for sulfate, organic material, metals, or other components can be compared directly to bulk composition measurements from instruments such as the AMS, SAGA, or a variety of offline analytical methods (see Sect. 4.1). Murphy et al. (2006) derived sulfate and organic mass fractions from PALMS negative polarity spectra by calibrating airborne data to a quadrupole AMS. New calibrations for sulfate and organic mass fraction were performed on positive mass spectra for

the current study using realistic atmospheric surrogate particles, shown in Fig. 7. Aqueous solutions were nebulized to generate a submicron aerosol population that was dried to RH<40% and sampled with PALMS. Solutions were composed of ammonium sulfate mixed with varying amounts of sulfuric acid, sucrose, and mixed dicarboxylic acids (see Table S1). Positive spectra are very similar to free tropospheric aerosol spectra in the SO particle class. Another set of calibration solutions contained ~1% potassium to mimic biomass burning aerosol.

25       Figure 7a shows the PALMS response as a function of aerosol organic mass fraction. The organic signal fraction, $sf_{org}$ defined as the intensity ratio of organic peaks/(organic + sulfate peaks), is fit to the organic mass fraction mforg using the functional form,

$$mf_{org} = \frac{m_{org}}{m_{org}+m_{sulf}} = \frac{sf_{org}}{\alpha+sf_{org}(1-\alpha)} \tag{2}$$

The single parameter $\alpha$ represents the relative ionization efficiency (RIE) of organic material to sulfate. No systematic differences were found in the PALMS response to organic and sulfate mixtures with and without potassium. The calibration fit is applied to

the signal fractions in Fig. 7b. The organic mass fraction for individual particles can have large errors due to the inherent particle-to-particle variability in SPMS spectra. However, errors are quickly reduced when averaging over a population of particles. To estimate precision uncertainty, particles were arranged into groups of increasing size, and the average organic mass fraction for each population was compared to the solution mass fraction, Fig. 7c. The relative standard deviation converged to 8% when averaging ≥15 spectra, and errors exhibited no trend with mass fraction (not shown). Coincident with PALMS, an Aerodyne AMS

using a quadrupole mass analyzer (Jayne et al., 2000) measured sulfate and organic mass of the generated aerosol. In Fig. 7b, average mass fractions from the AMS show similar deviations from the 1:1 line as the PALMS averages.

      The calibration is applied to positive spectra for all non-refractory particle classes, including SO, BB, MT, and FO. The organic or sulfate mass concentration for an individual particle class can be calculated by multiplying the mass concentration by



the organic or sulfate mass fraction. Similarly, total organic and sulfate mass concentrations are calculated as the sum of organic and sulfate mass concentrations from all non-refractory particle classes. These mass concentrations are conceptually comparable to bulk aerosol analysis of organic and sulfate made by common online (e.g., aerosol mass spectrometry) and offline (e.g., ion chromatography) techniques. By truncating the PALMS size range accordingly (Hu et al., 2017), direct comparisons between

PALMS and these other measurements can be made. Total uncertainties in PALMS sulfate and organic mass concentrations are estimated from uncertainties in the simplified size distribution (Sect. 3.3) combined with errors in mass fraction, OPS volume, particle classification, and particle density (see Appendix A). For ATom campaigns using 3-min sample periods, estimates of total relative uncertainties ($1\sigma$) are 40-50% for mass concentrations $\leq 0.01$ µg m$^{-3}$ and 20-35% at higher concentrations. Any potential biases due to extrapolation of accumulation mode composition to sizes below the PALMS size range (Sect. 3.3) are not included.

Primary sulfate on sea salt or mineral dust, as well as secondary sulfate accumulated on other particle types, is not included. The analysis assumes that dry aerosol mass is composed entirely or organic and sulfate material. Constituents like ammonium, nitrate, chloride, and alkali metals are disregarded. In most free tropospheric environments these components account for a small fraction of the aerosol mass. In the continental boundary layer or under polluted conditions, higher levels of ammonium and nitrate will introduce a high bias to the derived organic and sulfate mass concentrations. Future calibration studies can address

the aerosol nitrogen content.

## 4. In-flight performance

### 4.1 Comparison to other aerosol composition measurements

Figure 8 compares PALMS mass concentrations for sulfate and mineral dust with other online and offline techniques during routine airborne measurements. PALMS sulfate mass is calculated as the sum of all non-refractory particle types (SO, BB,

MT, and FO) and accounting for the sulfate mass fraction of each particle type (Sect. 3.7). PALMS sulfate mass concentration shows excellent agreement with other airborne sulfate measurements over several orders of magnitude (Fig. 8a). For the AMS comparison the PALMS and LAS size ranges are truncated using the AMS size-dependent lens transmission efficiency, which is similar to that reported in Hu et al. (2017).

There exist few standard methods to derive accurate mineral dust aerosol concentrations. The Interagency Monitoring of

Protected Visual Environments (IMPROVE) program performs routine aerosol composition measurements at over 200 ground sites throughout the US. Particles are collected on filters over a 24 hour period every few days. Bulk concentrations of aerosol components are measured using particle induced X-ray emission (PIXE), X-ray fluorescence (XRF), and other offline analytical techniques. Sulfate is derived from sulfur measurements assuming partial neutralization by ammonium. Soil dust concentration is derived from crustal metal concentrations and applying a basic mineralogy. In Fig. 8b and 8c, PALMS sulfate and mineral dust

mass concentrations are compared to IMPROVE data for airborne sampling in the continental boundary layer near an active IMPROVE site. PALMS airborne data are truncated to $D_a$ <2.5 um to match the IMPROVE size range. Spatial and temporal variability render this an indirect comparison. Nevertheless, the sulfate measurements are strongly correlated, suggesting that measurement colocation is reasonable and that for sulfate, the snapshot obtained during short airborne segments can often be representative of the daily average. Mineral dust is also positively correlated but exhibits higher variability than sulfate. The

variability is large compared to estimated uncertainties, suggesting that real atmospheric variability rather than measurement error is the cause. This is not surprising given the different sources of sulfate and mineral dust. Dust resuspension from land surfaces is a strong function of wind speed, and localized wind patterns give ground-level dust a high variability at small spatial scales, whereas secondary sulfate sources are more regional in scope. Additionally, the different size ranges for sulfate and dust can lead




to different loss rates due to precipitation scavenging or gravitational deposition. Despite the imperfect sampling overlap, the qualitative agreement and positive correlation in Fig. 8c furthers confidence in the ability of PALMS to measure absolute mineral dust mass concentrations.

## 4.2 Examples of mineral dust mass over the US

To demonstrate the utility of the new quantification method, we calculate the average mineral dust mass over the continental US. Figure 9 compares PALMS mineral dust mass concentrations for four airborne campaigns that span different regions and seasons. In general, the large majority of dust mass was present in the coarse mode, $D_{ve} > 1$ μm. For the summertime campaigns, concentrations decrease steadily with altitude, suggesting that dust was emitted from regional sources and removed during vertical transport. For the DC3 campaign the dust maximum occurred in the mid-troposphere. This profile is consistent
with Asian dust sources contributing significantly to springtime US dust loadings (Chin et al., 2007) and also to convective lofting of dust. The MACPEX campaign targeted large scale convective systems. The dust mass increase at 11-12 km is consistent with deep convective outflow. For the SEAC4RS, DC3, and NEAQS campaigns, the aircraft inlets and aerosol instrumentation measured sizes up to 4 μm and captured nearly the entire coarse mode. An exception was a weak Saharan dust plume encountered over the Gulf of Mexico, where external cloud-aerosol probes showed particles up to 20 μm.

## 15   5. Summary and recommendations

We present a new method to measure composition-resolved aerosol size distributions and quantitative concentrations using single-particle mass spectrometry (SPMS) combined with absolute particle concentration measurements. This method introduces a critical new capability for fast-response measurement of mineral dust aerosol concentration from aircraft platforms. Other common refractory and non-refractory particle concentrations are also determined, including sea salt, sulfate/organic internal mixtures,
biomass burning, heavy fuel oil products, as well as aerosol chemical components distributed across many particle types such as sulfate and organic material.
The principal strengths of this approach are summarized below:

- PALMS and many other SPMS instruments fundamentally classify individual aerosol particles into distinct composition types. PALMS detects all major particle types in the atmosphere, including refractory particles.
- Particle types and sub-types can be defined flexibly to suit a particular science objective, e.g., particles of stratospheric versus tropospheric origin. Definitions can be refined to characterize newly discovered particle types.
- Fast time response on the order of 1-5 mins for concentrations of ~10 ng m$^{-3}$ (see Appendix A).
- Intermittent clouds, plumes, or other events within a sample period can be excluded at high time resolution, e.g., 1 sec.
- Concentration products can be derived for many historical PALMS and other SPMS datasets that have coincident particle
size distribution measurements.
- The size distribution measurement can employ any of several standard sizing techniques ($D_{opt}$, $D_{mob}$, $D_a$), for which many commercial units are available.
- Stable SPMS detection efficiency is not a prerequisite. Variations in size-dependent detection efficiency due to different sampling conditions or instrument configurations only affect the product time resolution and uncertainty, not the derived
concentrations. The alternative method that scales observed SPMS detection rates by fixed detection efficiencies results in large, unquantifiable uncertainties in integrated number, surface, and volume.





PALMS differentiates externally mixed particle types based on mass spectral signatures. Climate-relevant particle types such as mineral dust, biomass burning, and sea salt are readily distinguished, giving PALMS and other SPMS instruments the unique capability to measure these important primary aerosol species with high time and size resolution. Quantification of rare types from unique emission sources is also possible, such as industrial metallic particles and bioaerosol. In addition to concentrations for

individual particle classes, calibration of mass spectral signal ratios allows for quantification of sub-components within a particle classes or across several classes. Aggregate sulfate and organic masses are determined here by summing the contributions over the non-refractory particle classes.

Time resolution and statistical accuracy for aircraft studies are primarily limited by data acquisition rate across the entire accumulation and coarse modes. Reducing native size resolution to 3 or 4 bins improves sampling statistics and allows for faster

time resolution, while introducing modest systematic errors (typically 5-25% in volume).

Particle densities and dynamic shape factors for each particle class are prescribed from literature or determined by simultaneous optical and aerodynamic size measurement in PALMS. Densities for particles types that are predominantly sulfate and organic material by mass are determined from their mass fraction. Density and shape values affect the size bin alignment between PALMS and the particle size spectrometer. Concentration products are more sensitive to prescribed density and shape than to assumptions

of particle refractive index, with the exception of highly absorbing species like BC. In agreement with limited literature studies, we find that the dynamic shape factor for irregular particles such as natural and synthetic mineral dust can be significantly higher under near-vacuum conditions than at atmospheric pressure.

We conclude with several recommendations for adopting this method for airborne SPMS measurements:

- SPMS users should minimally recreate Fig. 3, 4, 5, and 10, and also an OPS counting efficiency and sizing accuracy
assessment (Kupc et al., 2018), to help estimate the principal components of concentration uncertainty in their respective systems.

- User selectable parameters for particle type concentration products include definition of particle types, the sample averaging time, number and range of sizes bins, minimum number of mass spectra per size bin, and densities and shape factors for all particle types.

- Increasing the size range and improving detection efficiency of single-particle instruments across the size mode(s) of interest are the most important parameters to reducing biases, systematic errors, and statistical uncertainties that translate directly into faster time resolution.

- Aircraft inlets and size distribution instruments must demonstrate effective transmission and detection efficiency through the coarse mode, which in the background continental troposphere extends to $D_{ve} \geq 4$ µm and can be larger in plumes or
at low altitudes. In the presence of dust events or in the marine boundary layer, nearly all aircraft inlets will sample only a minor fraction of the coarse mode mass (Brock et al., 2019).

- Desired attributes for airborne optical particle spectrometers include a size range that covers the full accumulation mode (D~0.06-1 µm) and a large fraction of the coarse mode (D~1-10 µm), a sample flow rate of ~> 1 lpm to allow reasonable statistical sampling times for coarse aerosol but while limiting coincidence errors for small particles, and the ability to
operate with a large pressure difference between instrument interior and a pressurized aircraft cabin. The combination of a dedicated accumulation mode instrument with a separate coarse mode instrument operating at a higher sample flow is advantageous.

- Sampling lines can be actively dried to remove aerosol water. Otherwise, residual water must be accounted for in prescribing particle density and for OPS sizing when deriving dry volume and mass concentrations. Inlet RH should be





kept below 40% for sea salt, and preferably lower for sulfuric acid or sulfate internally mixed with organics. Operators should consider tradeoffs between active drying and possible loss of volatile aerosol material.

- Airborne sampling inside water and ice clouds produces a variety of artifact particles. Even brief cloud segments can perturb average concentrations by large factors, particularly for coarse mode and refractory aerosol, although submicron non-refractory measurements are also affected (Cziczo and Froyd, 2014; Murphy et al., 2004b; Weber et al., 1998). Artifact contributions to measured aerosol properties are difficult to predict, and measurements inside clouds using typical aerosol inlets should be considered suspect unless they have been thoroughly validated under specific cloud conditions.

**Appendix A. Uncertainty sources and limits of detection**

The principal uncertainty sources in deriving SPMS particle type concentrations are OPS counting and sizing errors, simplification of the size distribution (Sect. 3.3), and the statistical noise for detecting individual particle types within each size bin. Minor contributors to uncertainty include density and shape factor errors, provided they can be validated or constrained in the SPMS system, and particle classification errors. Estimation of total measurement uncertainty is not always a simple propagation of individual error sources because compensating factors can buffer some types of error. For example, an error in $D_{ve}$ due to inaccurate density or dynamic shape factor that that does not shift the particle to a different size bin would not contribute to any additional uncertainty in number, surface, and volume concentrations (Sect. 3.5). Another example is that poor SPMS particle statistics within a size bin will contribute a variable amount of uncertainty to total concentration, depending on the relative concentration within that bin (Sect. 3.3). Error sources vary for different SPMS instruments, the chosen methodology parameters (Sect. 4), and the sampling environment.

**A.1 Size distribution**

Overall OPS uncertainty is dominated by systematic uncertainties from the sample volume measurement, the error in prescribed refractive index, and counting statistics. Given reasonable constraints on refractive index, typical overall number, surface area, and volume uncertainties for the accumulation mode for 3 min samples are ~2, ~10, and ~15% for aerosol loadings of ≥0.1 um³ cm⁻³. In very clean environments (N<10 cm⁻³) or for supermicron sizes where concentrations are often very low (N<<1 cm⁻³), statistical sampling limitations are higher, and longer sampling times or regional averages are recommended to reduce statistical error. See relevant error analyses in Brock et al., 2011, 2019 and Kupc et al., 2018.

**A.2 Particle classification**

Particle classification error can contribute to particle type concentration uncertainty, but is typically lower than other error sources. Classification error is difficult to determine for all principal atmospheric particle types (Table 1) due to lack of accurate reference measurements under realistic atmospheric conditions, e.g., mineral dust and biomass burning. Laboratory experiments for surrogate particles can help estimate typical classification error. For a mixed sample of laboratory air and resuspended Arizona Test Dust (Power Technology, Inc), a compositionally diverse dust surrogate, a manual verification of 2500 particle spectra classified as dust showed <3% classification error. Errors can sometimes be estimated by sampling in atmospheric environments overwhelmingly dominated by one particle type, such as the remote marine boundary layer for coarse sea salt, thick dust plumes for coarse mineral dust, and thick smoke plumes for biomass burning particles. A manual inspection was performed of 1255 particle spectra with sizes $D_{ve}$>1 μm sampled during ATom-1 pristine MBL periods. PALMS classification routines identified 1094 sea salt spectra, for which 2 (0.2%) were false positives, and 9 sea salt spectra (0.8%) were mis-classified as other particle





types. Typical PALMS classification errors are <5% for all classes in Table 1 and represent a minor contribution to particle type concentration uncertainty. However, mis-classification of a common particle type as a rare particle type can contribute a larger relative uncertainty. For instance, if 10% of biomass burning particles were mis-classified as mineral dust during the SEAC4RS campaign, dust volume concentration would be anomalously high by 30%. SPMS users should consider possible biases that

systematically suppress the identification of a particular particle type due to low ion signal (e.g., sulfuric acid) or poor quality spectra (e.g., mineral dust).

**A.3 Density and shape factor**

Density and shape factor errors affect conversion to $D_{ve}$ and also the conversion of volume to mass concentration. Density uncertainties should be determined for each SPMS particle type and possibly for particle subcomponents such as sulfate and organic

material. Simultaneous optical and aerodynamic particle sizing (Sect. 3.5) or similar methods help constrain prescribed values. Estimated PALMS uncertainties are ±0.1 g cm-3 for non-refractory particle type densities, ±0.15 g cm-3 for dust and sea salt densities, and ±0.15 for the dynamic shape factor of mineral dust. The EC particle type is less well constrained in both density and shape. For common particle types, such as sulfate/organic/nitrate, when 100's of spectra are observed during a sample period, statistical sampling uncertainties are reduced to levels where density and dynamic shape factor errors can begin to compete. For

rare particle types, statistical uncertainties dominate (see below), and dynamic shape factor errors are typically minor contributions.

**A.4 Statistical uncertainties in volume concentrations**

In this section we consider the uncertainty in PALMS particle type volume concentrations due to statistical sampling limitations. Statistical uncertainties for each particle type can be estimated at every time point by assuming Poisson statistical behavior. Uncertainties for the particle number fractions for each class are determined in each size bin and are propagated through the

multiplication of number fractions by optical particle spectrometer concentrations. The total uncertainty in volume concentration, $\delta V_i$, for particle class $i$ across all diameter bins $d$ in a sampling period with $N_d$ total particles is determined as follows. The uncertainty for each particle class and size bin is based on the assumption of a Poisson probability distribution. To treat particle counts as independent variables we define $N_{j,d}$ as the number of counts for all non-$i$ classes.

$$N_d = N_{i,d} + N_{j,d} \tag{A1}$$

$$\delta N_{i,d} = \sqrt{N_{i,d}} \tag{A2}$$

$$\delta N_{j,d} = \sqrt{N_{j,d}} \tag{A3}$$

Particle classification errors would add to the count uncertainties in Eq. (A2) and (A3). The number fraction of particles in class $i$ and size bin $d$ is $f_{i,d}$ with an uncertainty $\delta f_{i,d}$ that is determined from error propagation formulae.

$$f_{i,d} = \frac{N_{i,d}}{N_{i,d} + N_{j,d}} \tag{A4}$$

$$\delta f_{i,d} \equiv \sqrt{\left(\left|\frac{\partial f_{i,d}}{\partial N_{i,d}}\right| \delta N_{i,d}\right)^2 + \left(\left|\frac{\partial f_{i,d}}{\partial N_{j,d}}\right| \delta N_{j,d}\right)^2} \tag{A5}$$

$$\delta f_{i,d} = \frac{\sqrt{\left(N_{j,d} \delta N_{i,d}\right)^2 + \left(N_{i,d} \delta N_{j,d}\right)^2}}{\left(N_{i,d} + N_{j,d}\right)^2} \tag{A6}$$

The uncertainty in the volume for each class and bin is $\delta V_{i,d}$, which is determined from the number fraction uncertainty and volume measured by the particle spectrometer, $V_d$. In order to limit this investigation to statistical uncertainties in PALMS particle class



concentrations, the uncertainty in the particle spectrometer measurement is ignored here, i.e., $\delta V_d \equiv 0$. In practice, users should include $\delta V_d$ in Eq. (A8) when estimating total concentration error.

$$V_{i,d} = f_{i,d} V_d \tag{A7}$$

$$\delta V_{i,d} = \sqrt{\left(\delta f_{i,d} V_d\right)^2 + \left(f_{i,d} \delta V_d\right)^2} = \delta f_{i,d} V_d \; ; \; (\delta V_d \equiv 0) \tag{A8}$$

$$\delta V_i = \sqrt{\sum_d (\delta V_{i,d})^2} \tag{A9}$$

To then calculate the mass concentration for a particle sub-component such as sulfate, $M_{sulf}$, (see Sect. 3.7), the mass concentration $M_i$ for class $i$ consisting of all non-refractory particles is determined using particle densities, $\rho_i$, and the average sulfate mass fraction, $mf_{sulf,i}$ is applied.

$$M_i = \rho_i V_i \tag{A10}$$

$$M_{sulf} = mf_{sulf,i} M_i \tag{A11}$$

$$\delta M_i = \sqrt{(\delta \rho_i V_i)^2 + (\rho_i \delta V_i)^2} \tag{A12}$$

$$\delta M_{sulf,i} = \sqrt{\left(\delta mf_{sulf,i} M_i\right)^2 + \left(mf_{sulf,i} \delta M_i\right)^2} \tag{A13}$$

Note that particle classification error between non-refractory particle types does not contribute any uncertainty to sulfate mass concentration if the particle classes all use the same calibration function (Fig. 7).

It should be noted that Poisson statistical errors for zero- and low-count samples can be problematic. The estimated standard deviation of 0% for zero-count samples is not realistic. Likewise, the 100% standard deviation for 1-count samples is often unsuitably large for rare particles types. This extreme variability of sample-to-sample error for rare particle types can render single-sample error estimates of little practical value. This problem becomes more prevalent with increasing number and specificity of user-defined particle classes. A possible alternative to deriving reasonable single-sample error estimates for rare

particle types is to determine a true mean and true standard deviation across multiple samples within a similar atmospheric environment. Despite these limitations, the Poisson model is self-consistent such that estimated errors propagated across multiple samples converge properly to the true overall error, and the standard Poisson model is presented here.

### A.5 Statistical analysis applied to PALMS airborne measurements

Figure A1a shows how statistical uncertainties for five particle classes vary with their volume concentration for one aircraft flight.

The flight was chosen to represent a variety of tropospheric environments and includes both very clean conditions and an MBL with a substantial coarse mode. Figure A1b summarizes statistical errors for coarse particle concentrations of mineral dust and sea salt, and for sulfate mass determined from non-refractory accumulation mode particles. Figure A1b also includes data from a second flight with elevated concentrations of upper tropospheric mineral dust. Minor particle types with few particles per sampling period have lower derived concentrations and higher relative uncertainty. At low volume concentrations of ~0.01 um³ cm⁻³ typical

(interquartile) errors for a 3-min sample are 50-80% for rare particle types such as dust and sea salt, and 10-30% for common particles such as sulfate/organic/nitrate. Relative errors decrease to 20-50% and 5-10%, respectively, for concentrations ≥0.01 um³ cm⁻³. Propagation of density and sulfate mass fraction errors result in typical sulfate mass concentration errors of 15-40% at 0.01 µg m⁻³ and 10-25% for higher concentrations. The convergence of PALMS concentration standard deviations toward 100% below ~0.01 um³ cm⁻³ is a consequence of low-count samples for rare classes, whereby many samples should be averaged to reduce

statistical noise. Figure A1c illustrates how statistical noise depends both on sample population and absolute abundance and is also a complex function of how particles observed by PALMS are distributed across the dominant size modes. For common



particle types that account for >30% of the total volume, the relative statistical error in a 3-min time period reduces to <30% when >50 particles are sampled in the dominant volume mode.

PALMS and other SPMS instruments have zero background at zero concentration. Since background subtraction is not required, the lower limit of detection (LLOD) depends only on the measurement uncertainty and can be estimated as 1.645 times the standard

deviation of a low concentration sample (Armbruster and Pry, 2008). Since the detection efficiency at the time of measurement can vary over an order of magnitude and is most sensitive to instrument inlet and laser alignment but also to ambient pressure, the actual LLOD throughout a measurement period can only be estimated.

A simple estimate of number concentration LLOD can be made from sampling counts and an average detection efficiency. The minimum detectable concentration depends on particle size since the PALMS detection efficiency increases from ~$10^{-4}$ for particles

150 nm in diameter to ~0.05 for 500 nm particles to ~0.1 for supermicron sizes (see Fig. 6). The peak of the PALMS size distribution is typically about $D_{ve}$ =400 nm. In a one-minute sample period, detection of one 400 nm particle corresponds to a concentration of approximately 0.07 cm$^{-3}$, with an LLOD of ~0.1 cm$^{-3}$. For $D_{ve}$ = 150 nm the LLOD is ~20 cm$^{-3}$.

Estimation of LLOD for the derived concentration products is more complex. The particle volume LLOD depends on both the PALMS size response and the shape of the volume size distribution. The examples in Fig. A1 show that typical uncertainties in

particle volume are still quantifiable at very low atmospheric concentrations. Although LLODs can be estimated from the uncertainties in Fig. A1, the statistical analysis does not provide an obvious volume below which quantification is not feasible, nor can a single LLOD value be derived. The statistical analysis (Fig. A1c) does however provide a good rule of thumb that a minimum of 50 total spectra are needed to generate volume concentrations for the dominant particle types within acceptable statistical noise of around <30% using 3 size bins. Fig. A1d shows the sampling time required to measure 50 particles in the free troposphere.

Under typical lower tropospheric conditions with an accumulation mode volume of ~1 μm$^3$ cm$^{-3}$, PALMS can quantify the dominant particle types in <1 min of sampling. In very clean conditions of 0.01 μm$^3$ cm$^{-3}$ occasionally observed in the upper troposphere, PALMS needs a median sampling time of 1 minute. Fig. A1a and A1b show that at concentrations below ~0.01 μm$^3$ cm$^{-3}$ statistical uncertainties increase toward 100%. These considerations suggest a reasonable estimate of LLOD as 0.01 μm$^3$ cm$^{-3}$ in one minute of sampling for dominant particle types. On the other hand, particle types with low relative abundance may require

thousands of sampled spectra to determine their concentration to an uncertainty of <30%. Since the derived concentration is also based on size distribution measurements, the optical particle spectrometer sample flow and detection efficiency could potentially contribute to the detection lower limit. However, most particle sizing instruments have a LLOD that is similar to or better than PALMS.

**Appendix B. Shape factors for mineral dust**

We briefly discuss evidence to support large mineral dust shape factors observed by PALMS. Experimental studies of irregular particles including PSL aggregates, soot aggregates, and quartz particles demonstrate that $\chi_v > \chi_t$, and can $\chi_v$ can often approach ~$\chi_t^2$ (Alexander et al., 2016; Zelenyuk et al., 2006b). This higher relative drag force for irregular particles under free molecular flow is consistent with theoretical treatments by Dahneke (1973a, 1973b) for idealized shapes such as cubes and spheroids. However, no simple general relationship exists between $\chi_t$ and $\chi_v$ for realistic particles because the increase in particle drag in free

molecular conditions is a complex function of particle physical shape and surface morphology. For example, oblate and prolate spheroids behave differently, and concave surface features were not considered by Dahneke. Furthermore, when particle Reynolds numbers exceed ~0.1, particles begin to partially align with accelerating flows, increasing their apparent shape factor (Dahneke, 1973a; Kulkarni et al., 2011).

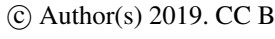



Following previous SPMS studies (Alexander et al., 2016; Zelenyuk et al., 2006b), a laboratory experiment was performed to investigate the large apparent $\chi_v$ for dust. A differential mobility analyzer was used to select commercial illite NX particles (Arginotec) with a fixed mobility size, $D_{mob}$, and the distribution of aerodynamic sizes was measured by PALMS. Given the material density of 2.65 g cm⁻³ and shape factor at conditions of the DMA of $\chi_t$(~1 atm) = 1.49±0.12 (Hiranuma et al., 2015), one

can derive the free molecular shape factor $\chi_v$ from theory (DeCarlo et al., 2004; Zelenyuk et al., 2006b),

$$\frac{d_{va}}{d_{mob}} = \frac{\rho_p}{\rho_0} \frac{1}{\chi_t \chi_v} \frac{C_c(d_{va}\chi_v\rho_0/\rho_p)}{C_c(d_{mob})} \tag{B1}$$

where $C_c$ is the Cunningham slip correction factor at DMA conditions. For particles with $D_{mob}$=0.580 µm the PALMS most probable aerodynamic diameter was $D_a$ = 0.485 µm, which when adjusted slightly to free molecular flow gives $D_{va}$ = 0.500 µm. This yields an average shape factor of $\chi_v$=2.27, which is significantly higher than $\chi_t$. A second method to derive $\chi_v$ uses a

parameterization of $\chi_t\chi_v$ for a variety of irregularly shaped particles (Zelenyuk et al., 2006b). Using equation 11 of that reference gives an even larger value, $\chi_v$ = 2.58. A third independent method uses the internal consistency of PALMS optical and aerodynamic diameters (Fig. 5b), which gives approximately $\chi_v$ =2.3 for illite NX. Ambient mineral dust particles sampled during the lab study had lower derived shape factors of $\chi_v$=1.7, which is comparable to values derived from dust in airborne studies but is still higher than typical transitional shape factors for dust ($\chi_t$ ~1.4, Davies, 1979; Kulkarni et al., 2011; Linke et al., 2006). These experiments

confirm that for dust particles, the shape factor near free molecular flow can be significantly greater than in other flow regimes, i.e., $\chi_v > \chi_t \sim \chi_c$.

**Data Availability**

Data is publically accessible from the NASA Distributed Active Archive Center (https://daac.ornl.gov/ATOM/campaign/) and the NASA Airborne Science Data for Atmospheric Composition (https://www-air.larc.nasa.gov/).

**Author Contributions**

KDF designed the study and performed airborne aerosol composition and microphysical measurements. DMM, PCJ, JED, JLJ, and GPS, performed airborne aerosol composition measurements. CAB, AK, KLT, CJW, JCW, and LDZ performed airborne aerosol microphysical measurements. GPS and AMM performed laboratory aerosol composition calibrations.

**Competing Interests**

The authors declare that they have no conflict of interest.

**Acknowledgements**

PALMS is supported by NOAA internal climate funding and also in part by NASA awards NNH12AT29I and NNH15AB12I. PCJ and JLJ were supported by NASA awards NNX15AH33A, NNX15AT96G, 80NSSC19K0124, and 80NSSC18K0630. The authors gratefully acknowledge the following people for their valuable input: Jin Liao, Daniel Cziczo, Troy Thornberry, Paula

Hudson, and David Thomson.



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



**Table 1.** PALMS principal atmospheric particle types

| Particle Type (abbreviation) | Description | Particle density[a] | Dynamic shape factor[b] | Predominant size mode | Typical abundance within size mode[c] | Principal positive spectral signatures | Dominant composition and notes |
|---|---|---|---|---|---|---|---|
| Sulfate/Organic/Nitrate (SO) | Internal mixture dominated by of sulfate, organic, nitrate material | 1.3–1.8[†] | 1 | accumulation | 50–90% | $C^+$, $C_2^+$, $CO^+$, $SO^+$, $H_xSO_y^+$, $NH_4^+$, $NO^+$, organic fragments | Typical background tropospheric particles from a variety of primary and secondary sources. Sulfate can be acidic or neutralized. The SO and BB classes contain the vast majority of sulfate and organic aerosol mass. |
| Biomass Burning (BB) | Emissions from biomass and biofuel burning. | 1.25–1.5[†] | 1 | accumulation | 10–30% | $K^+$, $C^+$, $C_2^+$, $CO^+$, organic fragments | Mostly primary organic carbon by mass and can contain internally mixed EC, Na, K, trace metals, chloride, and secondary material. |
| Elemental Carbon (EC) | EC that is not heavily coated by OC | 1.8 | 2.0 | accumulation | <1% | $C_x^+$, $C_xH^+$ | EC signatures are hidden by heavy organic coatings, whereby the particle is instead classified as SO. |
| Sea Salt (SS) | Sodium-rich particles emitted from oceans | 1.45[‡]; 1.8[‡] when heated | 1; 1.08 when heated | coarse with minor accumulation | <1% (30–100% MBL) | $Na^+$, $Ca^+$, $K^+$, $Na_2Cl^+$, $Sr^+$ | Also includes sodium-rich particles emitted by dry lake beds. |
| Mineral Dust (MD) | Crustal material | 2.5 | 1.6–1.8[‡] | coarse with minor accumulation | 5–50% | $Al^+$, $Si^+$, $K^+$, $Fe^+$, $Ca^+$, $Ba^+$, $Rb^+$, $Li^+$ | Includes a variety of pure and mixed mineralogies. |
| Meteoric (MT) | Ablated, recondensed meteoric material from the stratosphere that has accumulated sulfate | 1.5–1.7[†] | 1 | accumulation | 1% (5–40% strat) | $Fe^+$, $Ni^+$, $Mg^+$, $SO^+$, $H_xSO_y^+$ | Mostly sulfuric acid by mass until crossing below the tropopause. Metallic mass contribution is minor. |

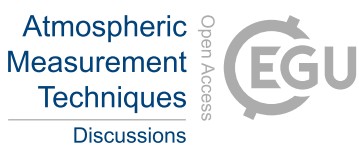

| Category | Description | Density[a] | χ_v[b] | Size mode[c] | Mass fraction | Ions | Description |
|---|---|---|---|---|---|---|---|
| Alkali Salt (KS) | Potassium-rich particles from continental emissions | 1.5[‡] | 1 | accumulation | <1% | $K^+$, $Na^+$, $Li^+$, $Rb^+$ | Potassium- and sodium-rich particles, often with other alkalis but low organic carbon and without typical crustal material. |
| Heavy Fuel Oil Combustion (FO) | Vanadium-rich particles unique to heavy fuel oil combustion | 1.3-1.6[†] | 1 | accumulation | 1% | $V^+$, $VO^+$, $C^+$, $C_2^+$, organic fragments | Mostly organic carbon and sulfate by mass with trace metals. Typically from ship emissions. |
| Other (OT) | Multiple particle types that are not classified in the above categories | 1.4 | 1 | accumulation and coarse | 5-10% | various | A variety of identified and unknown particle types, including biological, metallic, pyridinium, and others. |

[a] Density at measurement conditions

[b] $\chi_v$, within or near free molecular flow

[c] Mass fraction within the dominant size mode for background tropospheric air. Strat=lower stratosphere, MBL=marine boundary layer

[†] Calculated based on organic-to-sulfate mass ratio

[‡] derived from optical-aerodynamic analysis




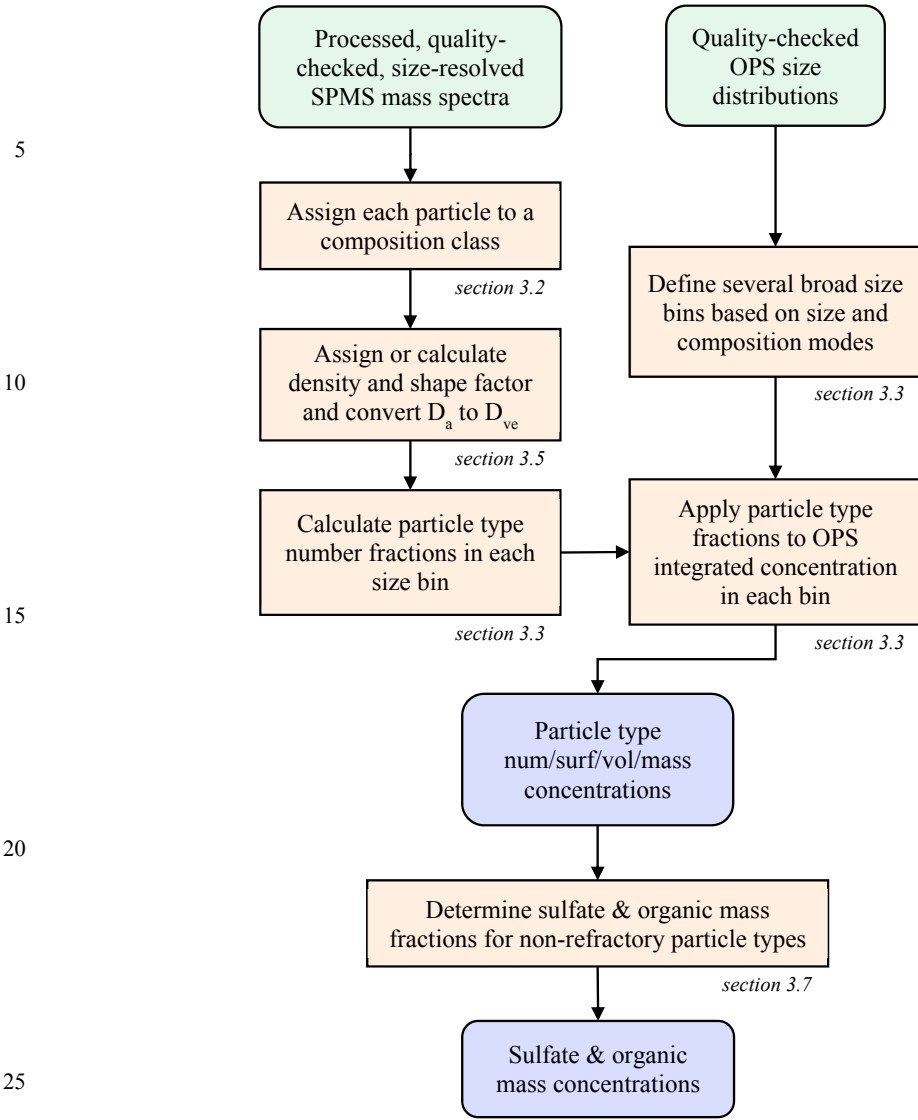

**Figure 1:** Flow chart to derive particle type concentrations and bulk sulfate and organic mass concentrations from SPMS and OPS data.



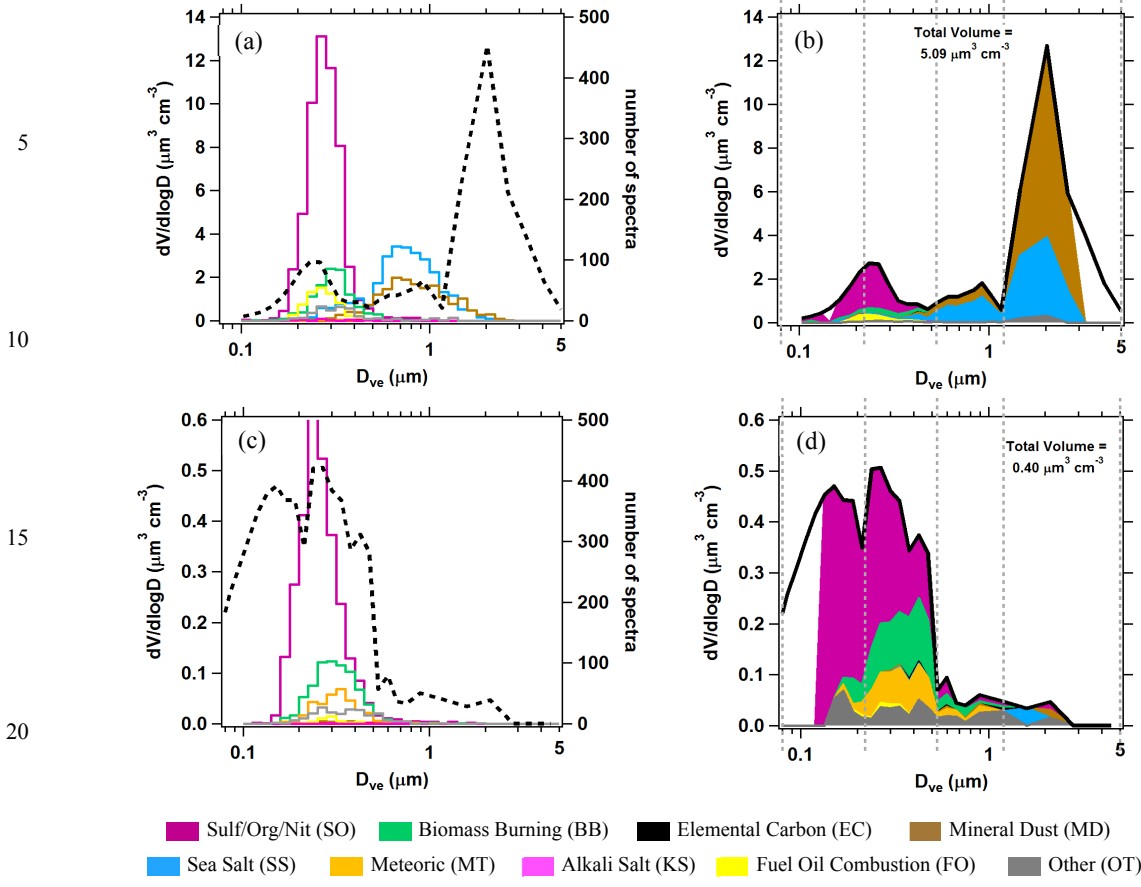

**Figure 2:** Quantification of PALMS particle classes. (a,c) Raw PALMS counts for different particle classes (colors) overlaid on the aerosol volume size distribution (dashed black). (b,d) For each diameter bin, the fractional contributions of each PALMS particle class are applied to the total volume. Vertical dashed lines define four broad diameter bins that are used to generate concentration products at higher time resolution (see text). The upper panels are from 39 min of sampling at low altitude over the Gulf of Mexico and contain influences from the marine boundary layer, long range mineral dust transport, and lower tropospheric pollution. The bottom panels represent the remote upper troposphere with minor influences from continental and stratospheric sources, sampled over 27 min.



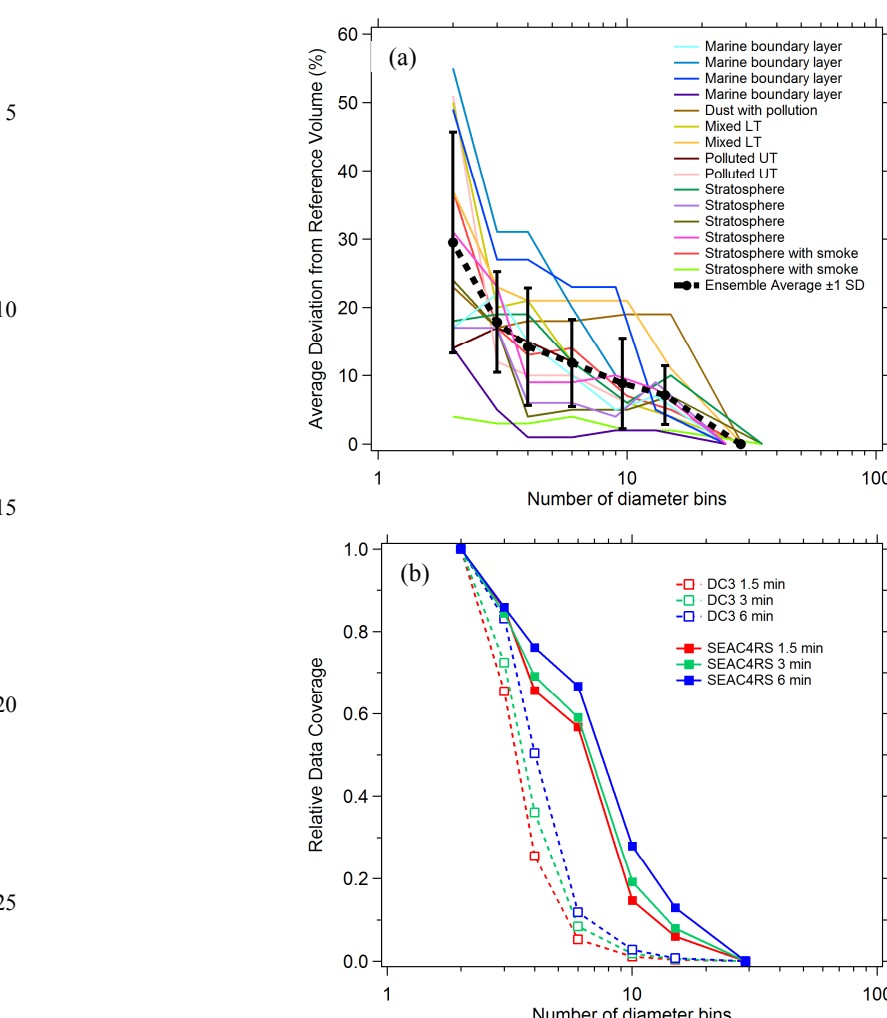

30 **Figure 3**. (a) Error in integrated volume introduced by reducing the size resolution of the analysis. Integrated volume was calculated for every PALMS particle class over several long flight segments with externally mixed but constant composition. Volumes computed at full diameter resolution (25-29 bins) provide a reference, and the average deviation for populous particle classes (contributing >5% of volume) is plotted as bins are combined. (b) Lines show the relative data coverage, defined as the number of time periods with >5 particles in every diameter bin, using three different raw sampling times for two flight campaigns.





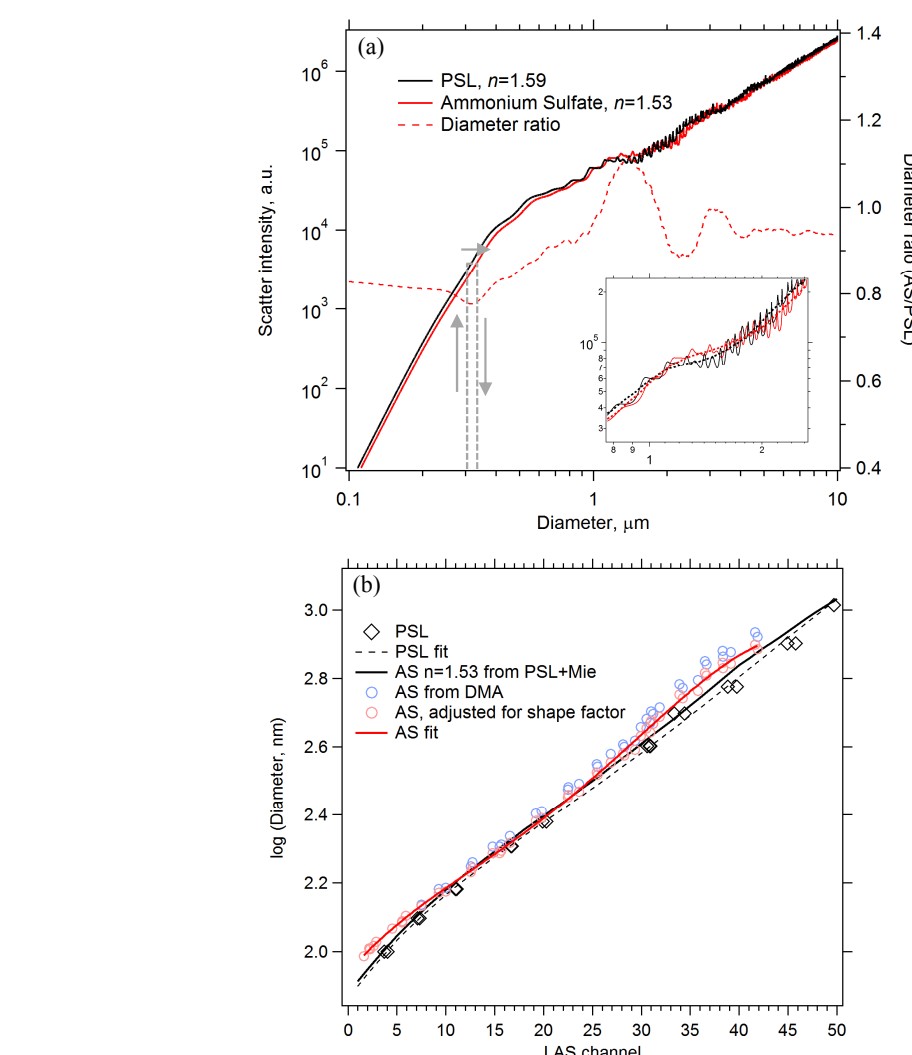

**Figure 4.** Using Mie theory to translate the response of an optical particle spectrometer between particles of different refractive index. (a) Calculated scatter intensities at λ=663 nm for PSL and effluoresced ammonium sulfate (AS) particles in the LAS instrument. PSL diameters are translated into ammonium sulfate diameters of the same scatter intensity using the Mie response curves, shown conceptually with grey arrows. The diameter ratio is plotted on the right axis. Smoothing must be applied to the Mie curves (inset a, dashed) in order to yield unique diameter translations. (b) A calibration curve derived from PSL particle standards (dashed black) is translated into an ammonium sulfate calibration curve (solid black) using Mie theory. Size-selected AS particles yield a directly measured AS calibration curve (solid red) to compare with the PSL-derived AS calibration curve.





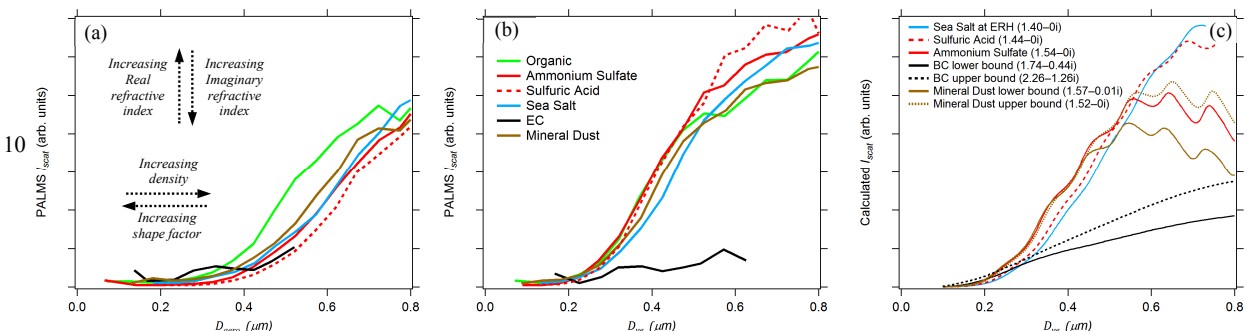

**Figure 5.** PALMS simultaneous optical and aerodynamic diameter measurements during the SEAC4RS airborne campaign. (a) Raw scatter intensities versus measured aerodynamic diameters for populations of different particle classes. Curves toward the right side represent particle with higher density/shape factor ratios, and a large real refractive index shifts curves upwards for these sizes. Sub-populations of the sulfate/organic/nitrate particle class are plotted for nearly pure (mass fraction>0.9) organic (green) and sulfate (red) particles. Lines are
20 the average of 100-72000 particle measurements. (b) Aerodynamic diameters are converted to volume-equivalent diameters by prescribing density and shape factors to each particle. Divergence at D>0.5 μm is due to Mie resonances, which are highly sensitive to refractive index. (c) Calculated Mie scattering intensities at λ=405 nm are plotted for different refractive indices (n-ki) that correspond to composition classes.





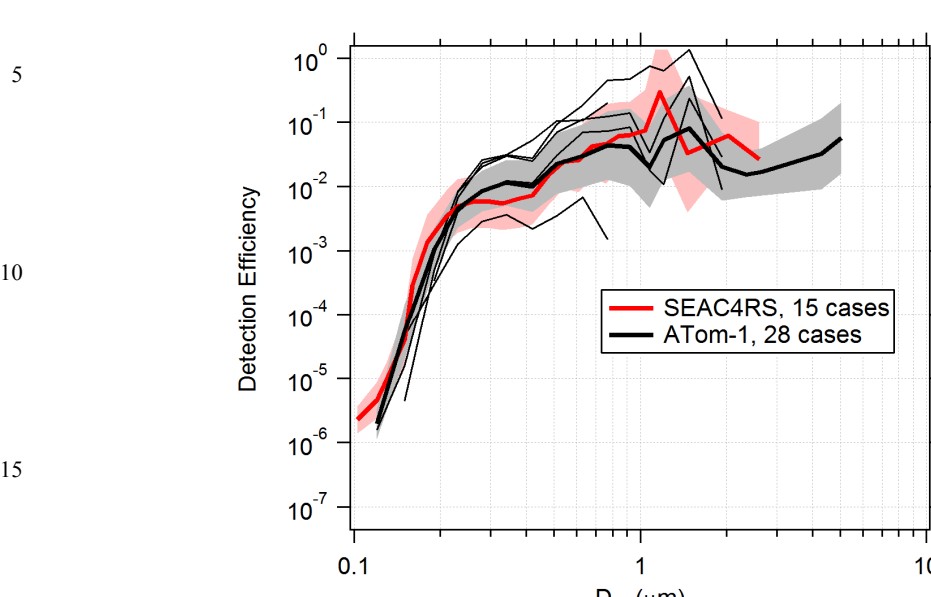

**Figure 6.** Detection efficiency curves for the PALMS instrument in flight. Detection efficiency was calculated for several cases, identified as clean flight segments when the particle data rate was not actively limited by software or hardware. Thick lines and shading are geometric means and standard deviations over all cases. Higher efficiencies for $D_{ve}$>3 µm during ATom are partly due to addition of a virtual impactor upstream of PALMS. Thin black lines are five individual cases during one ATom-1 flight. Variations in altitude, particle composition, inlet performance, and unknown factors result in detection efficiencies that vary by >x10 within and between flights. The lowest thin line was a case where a buildup of aerosol material on the pressure reduction orifice altered particle trajectories inside the inlet.





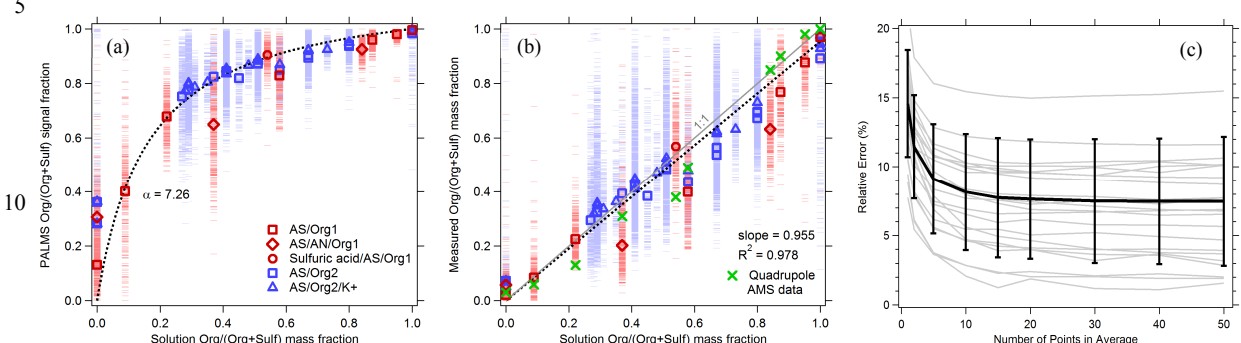

**Figure 7.** PALMS calibration of organic mass fraction for atmospheric aerosol surrogates composed of neutralized and acidic sulfate mixed with a variety of organic compounds (see Table S1). Blue points are ammonium sulfate/sucrose/adipic acid solutions, and red points are ammonium sulfate/sulfuric acid/dicarboxylic acid solutions. (a) Raw signal fractions (dashes) are fit to the solution organic mass fraction data (dotted line). Symbols are averages of 130-1900 spectra. (b) Calibrated organic mass fractions from PALMS (red and blue) confirm a linear response, with averages that exhibit similar deviations as a quadrupole AMS. The dotted line is a linear fit to PALMS data forced through zero. (c) Relative error (standard deviation/mean) for all calibration points as a function of the population size used to calculate the average.



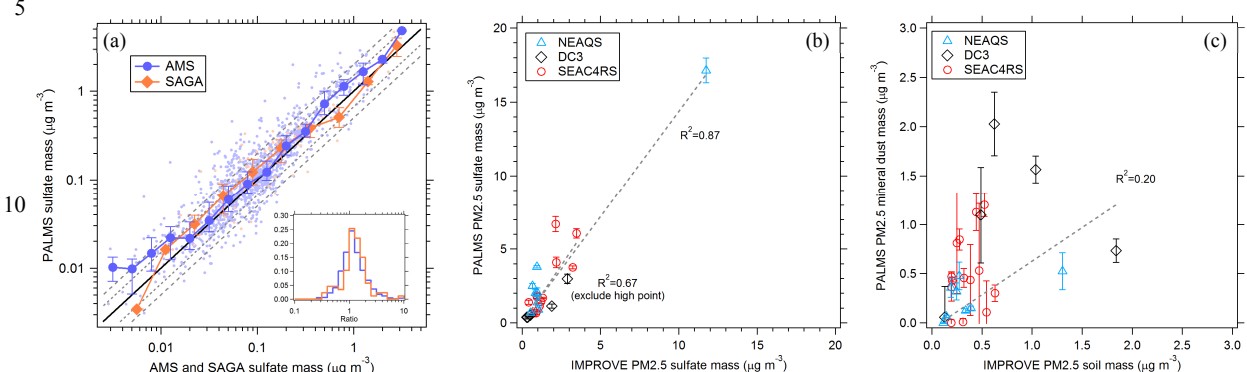

**Figure 8.** Comparison of PALMS with other speciated aerosol mass measurements. (a) PALMS derived sulfate mass at 3 min time resolution compared to co-located AMS and SAGA filter samples during the ATom-1 airborne campaign. Sea salt sulfate ($0.25 \times Na^+$) is subtracted from SAGA data. SAGA filters taken over an altitude range >3 km are excluded. Small points are 3-min averages (blue) or represent one SAGA filter measurement (orange). Large symbols are medians with interquartile error bars. The solid black line is 1:1, and grey dashed lines are 1:1.5 and 1:2. The inset graph is a histogram of PALMS ratio to AMS or SAGA for all individual samples. PALMS sulfate (b) and mineral dust (c) mass are compared to nearby IMPROVE ground station data for three airborne campaigns. Each point is the average of airborne data for non-targeted flight segments in the continental boundary layer within 0.5° latitude and 1° longitude of an IMPROVE site that reported data that day. IMPROVE data are 24 h averages, and airborne segments are typically ~3-30 min duration during daytime. Error bars are estimated statistical uncertainty calculated as described in Appendix A. Dashed lines are weighted linear fits.





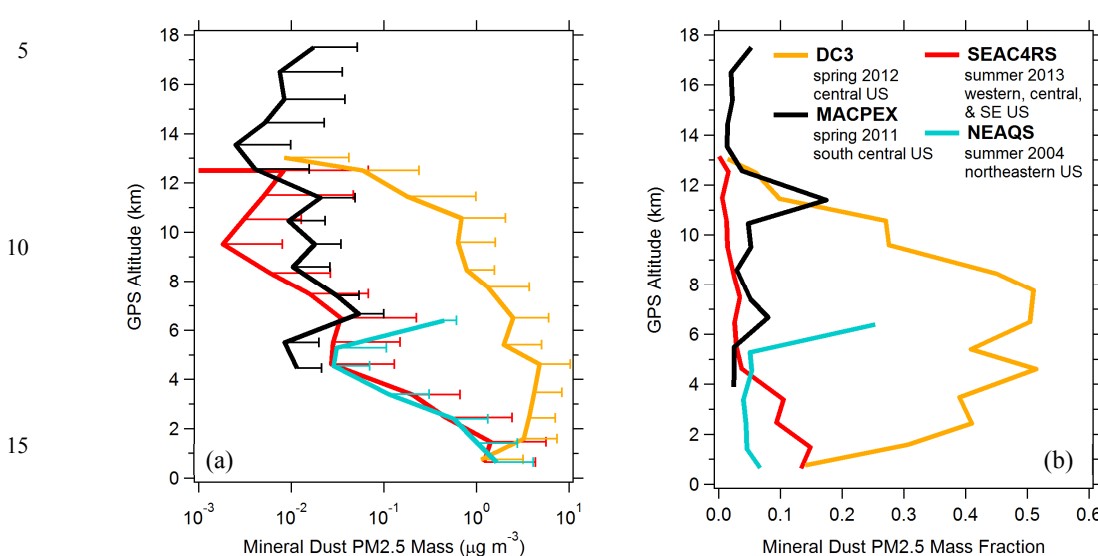

**Figure 9.** Vertical profiles of PALMS mineral dust mass (a) and mass fraction (b) over the continental US for $D_{ve}>0.1$ µm. Lines are campaign average concentrations binned at 1 km intervals from products generated at native resolutions of 3 min (SEAC4RS, DC3, NEAQS) or 5 min (MACPEX). Biomass burning plumes and clouds are excluded. By truncating the size range using a typical cyclone impactor transmission curve with $D_a(50\%)=2.5$ µm (http://www.urgcorp.com/), these mass concentrations are equivalent to a PM2.5 measurement. The MACPEX size range is limited by the optical particle spectrometer to $D_{ve}<1.5$ µm. Positive error bars are one standard deviation.





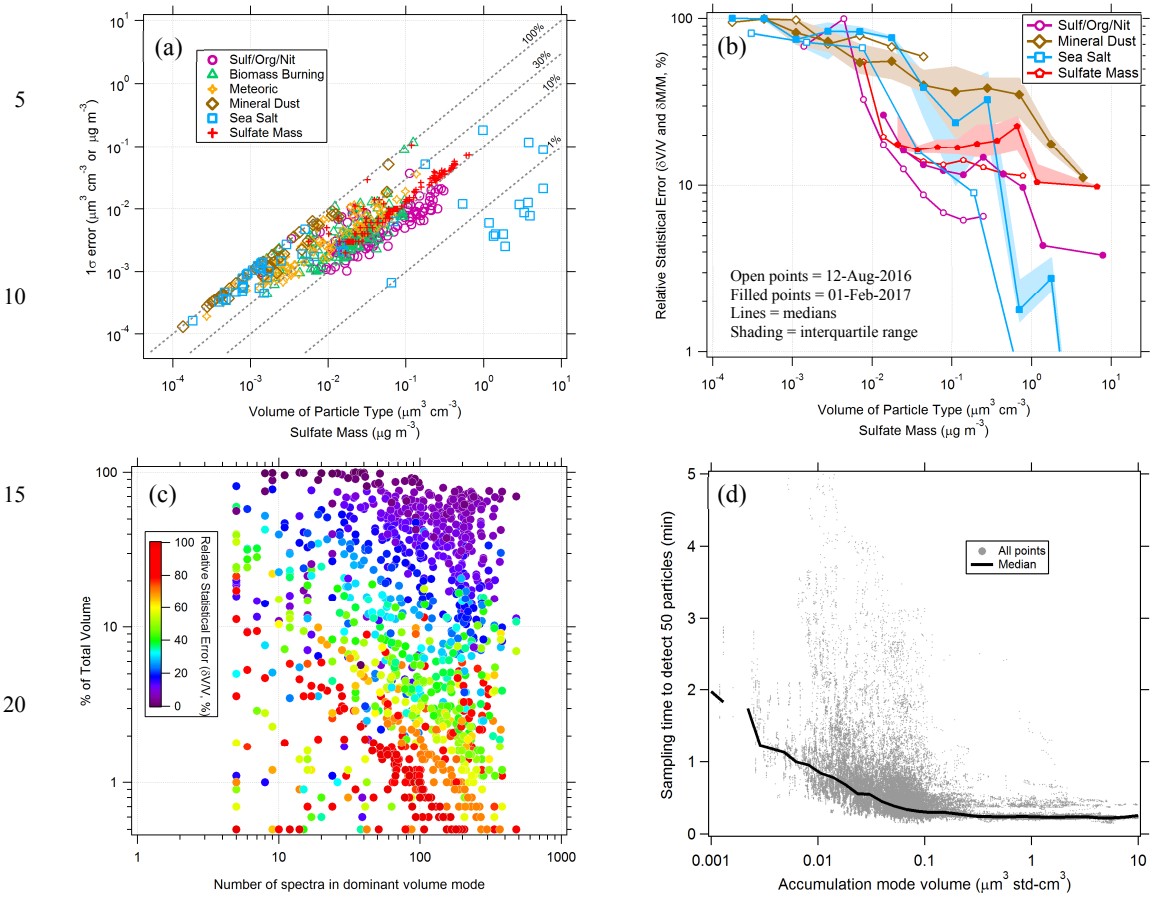

**Figure A1.** (a) Estimates of statistical uncertainties from the 12-Aug-2016 ATom flight for common particle types (sulf/org/nitrate, biomass burning, and MBL sea salt) and rare particle types (dust, meteoric, sea salt outside the MBL). Each point represents one 3-min measurement. (b) Relative errors for two flights are plotted versus volume concentration of each particle type. Errors for sulfate mass concentrations in (a) and (b) also include errors propagated from particle density and sulfate mass fraction. (c) The variation of statistical error with volume contribution and the number of analyzed spectra for 3 min time periods during the two flights in (b). (d) Analysis of the airborne sampling time necessary to acquire 50 particle spectra. Points are from three diverse flights in the free troposphere (>3 km), and the line is the median.