# Peer review of "A new method to quantify mineral dust and other aerosol species from aircraft platforms using single particle mass spectrometry"

_Atmospheric Measurement Techniques, 2019_

## Referee Comment (RC1) · Nicholas Marsden (Referee) · 10 Jul 2019

In this manuscript, the authors present a method for deriving size and composition resolved absolute particle concentrations from an aircraft based single-particle mass spectrometer by scaling particle classes to absolute concentrations measured by particle size spectrometers. Sulfate and organic concentrations in non-refractory aerosols are calculated with respect to lab generated standards. Whilst this approach is not new, the method offers enhanced temporal resolution of the measurement by placing single particle data into carefully selected size bins to enhance the particle number statistics. This is likely to be of interest to the community using aircraft based SPMS to measure

particles in low abundance, and has suitable content for a technical journal.

The benefit of the scaling method and of reduced size resolution are for correcting the total particle numbers are clear, but I do have some major concerns about particle detection bias that may skew the reported number fractions and therefore create large errors in the reported mass concentrations of compositional classes. This source of uncertainty needs to be discussed more thoroughly, particularly if the method is to be transferred to other aircraft based SPMS platforms. The current literature should be better referenced; particularly regarding the case for applying chemically resolved detection efficiencies in addition to size resolved scaling.

In my opinion, the manuscript will be worthy of publication once the potential instrument bias towards composition is more clearly discussed and it is made clear that these errors will be scaled into the absolute concentrations with the number fractions.

Please see the supplement for major and minor comments.

Please also note the supplement to this comment:
https://www.atmos-meas-tech-discuss.net/amt-2019-165/amt-2019-165-RC1-supplement.pdf

**Supplement:**

**A new method to quantify mineral dust and other aerosol species from aircraft platforms using single particle mass spectrometry**

**Major Comments:**

Previous attempts to scale SPMS data to tandem measurements should be describe in more detail in the introduction and in section 3.6. The authors described the work of Qin et al., (2006) as an attempt to scale SPMS data with pre-defined quantitative concentration obtained in controlled conditions (P13L26) , when it was in fact scaled to co-located APS measurements (with an hourly average). The same approach was used more recently to obtain mass concentration with the ATOFMS (Gunsch et al., 2018) which should also be referenced as an example of obtaining absolute concentration by size scaling. On the other hand, Marsden et al., (2016) compared data rates from a LAAPTOF with an APS and concluded an over-estimation in the relative fraction of sea-spray aerosol with respect to mineral dust due to instrument function and did not attempt to derive absolute concentration because of unquantified uncertainties. Subsequently, Shen et al., (2018) has defined composition resolved overall detection efficiency (ODE) for the LAAPTOF and has demonstrated that number fraction of compositional classes in ambient data could be incorrect by an order of magnitude in the uncorrected data (Shen et al., 2019).

Although all the above studies involved ground data, the same principals would apply to aircraft versions of these instruments. It maybe the unique design of the PALMS instrument does not suffer from such composition bias, but no evidence is offered in this manuscript. The chemical bias is described here as 'minimal' (P4L29) due to the particle hit rate of 90%, but does this apply to all particle types at all sizes? There is no reference to data. This discussion is necessary if the method is transferred to other instrument designs as the number fractions bias will be scaled to the absolute concentration. Marsden et al., (2018) showed that the spectral hit rate of mass selected mineral dust can vary by a factor of 2 due to mineralogy (not size transmission), therefore size scaling alone is insufficient. The absorbing characteristics of the particle at the ablation wavelength is an important factor. This should be discussed in the introduction, method and conclusion.

Another area of concern is the lack of discussion surrounding particle classification techniques and errors that arise due to matrix effects and complex particle mixing state. The PALMS system has a well-established classification scheme that need not be reviewed in detail here, but some acknowledgement of different result produced by different techniques should be offered with respect to the literature, particularly regarding ion mode of the TOFMS.

Hatch et al., (2014) showed that particle ageing can affect ATOFMS hit-rates and signal fractions in spectra of ambient particles. The core shell structure of coated particles was an important consideration when interpreting instrument response to organics and sulfate . I think this work should be referenced in Section 3.7 and some discussion of how well laboratory generated proxies represents internally mixed ambient particles using the relative ionisation efficiency method. Is a binary system (organic vs sulfate) sufficient in ambient particles with complex mixing state and structure? Marsden et al., (2019) argued for a ternary system with internally mixed dust. Also, a comparison of organic concentrations is missing in section 4.1 (A comparison of sulfate is given). This

important as it was the comparison organic comparison with AMS that was the most uncertain in Hatch et al., (2014).

Finally, please consider revising the structure of the methods section. For example, is a detailed description of the campaigns necessary for the understanding of this method? Would a table featuring the important parameters be sufficient? A summary paragraph under the section 2 main header would be helpful.

**Minor Comments:**

P1L20   The abstract could be consolidated, the particle types/classes are partially repeated from the paragraph above.

P1L34    The use of a virtual impactor to enhance sampling statistics is not demonstrated or discussed in any depth. Please add a section on this or remove form the abstract.

P2L13   This sentence could suggest (to a non-expert) that a single particle is ionised.

P2L30   Shen (2019) does discuss uncertainties in derived concentrations and also includes mineral dust.

P2L35    There are several topics of discussion in this paragraph that do not sit well together.

P3L18   There is a jump between mass concentration measurement and particle number counting in this paragraph that makes it a little incoherent.

P4L7     What particle size spectrometers? Not yet introduced.

P4 L10   The AMS instrument has not been introduced or defined.

P4 L11   SAGA inlet filters has not been introduced or defined.

P4L13   There is switch to a different campaign mid-paragraph.

P4L29   Is there a data or a reference for the hit-rate performance? How minimal is the chemical bias on the PALMS system.

P4L33   A summary of the post-processing method would be useful here.

P5L31   "The atmosphere consists of an external mixture of particle types" Is this conclusion or an argument? Maybe reword this sentence.

P9L26   Can you give a reference for the abundance of pure hematite in the atmosphere?

P14L25 Is the organic signal fraction calculated from peak area? Which peaks were used to do the calculation?

P15L10 An explanation of why mixing with sea-salt and mineral dust is not included. Potential matrix effects?

P16L5   I find the description of the mineral dust concentration (section 4.2) rather brief for a paper that has mineral dust in the title.

P16L15 The opening paragraph to the summary should be specific about what is new about the method i.e. the integrated size bins.

P17L14 What are concentration products?

P17L18 Not sure what are recommendation and what are general conclusions in this list.

**References**

Gunsch, M. J., May, N. W., Wen, M., Bottenus, C., Gardner, D. J., Vanreken, T. M., Bertman, S. B., Hopke, P. K., Ault, A. P. and Pratt, K. A.: Ubiquitous influence of wildfire emissions and secondary organic aerosol on summertime atmospheric aerosol in the forested Great Lakes region, Atmos. Chem. Phys., 18(5), 3701–3715, doi:10.5194/acp-18-3701-2018, 2018.

Hatch, L. E., Pratt, K. a., Huffman, J. A., Jimenez, J. L. and Prather, K. a.: Impacts of Aerosol Aging on Laser Desorption/Ionization in Single-Particle Mass Spectrometers, Aerosol Sci. Technol., 48(10), 1050–1058, doi:10.1080/02786826.2014.955907, 2014.

Marsden, N., Flynn, M. J., Taylor, J. W., Allan, J. D. and Coe, H.: Evaluating the influence of laser wavelength and detection stage geometry on optical detection efficiency in a single-particle mass spectrometer, Atmos. Meas. Tech., 9(12), 6051–6068, doi:10.5194/amt-9-6051-2016, 2016.

Marsden, N. A., Flynn, M. J., Allan, J. D. and Coe, H.: Online differentiation of mineral phase in aerosol particles by ion formation mechanism using a LAAP-TOF single-particle mass spectrometer, Atmos. Meas. Tech., 11(1), 195–213, doi:10.5194/amt-11-195-2018, 2018.

Marsden, N. A., Ullrich, R., Möhler, O., Eriksen Hammer, S., Kandler, K., Cui, Z., Williams, P. I., Flynn, M. J., Liu, D., Allan, J. D. and Coe, H.: Mineralogy and mixing state of north African mineral dust by online single-particle mass spectrometry, Atmos. Chem. Phys., 19(4), 2259–2281, doi:10.5194/acp-19-2259-2019, 2019.

Qin, X., Bhave, P. V. and Prather, K. A.: Comparison of two methods for obtaining quantitative mass concentrations from aerosol time-of-flight mass spectrometry measurements, Anal. Chem., 78(17), 6169–6178, doi:10.1021/ac060395q, 2006.

Shen, X., Ramisetty, R., Mohr, C., Huang, W., Leisner, T. and Saathoff, H.: Laser ablation aerosol particle time-of-flight mass spectrometer (LAAPTOF): Performance, reference spectra and classification of atmospheric samples, Atmos. Meas. Tech., 11(4), 2325–2343, doi:10.5194/amt-11-2325-2018, 2018.

Shen, X., Saathoff, H., Huang, W., Mohr, C., Ramisetty, R. and Leisner, T.: Understanding atmospheric aerosol particles with improved particle identification and quantification by single-particle mass spectrometry, Atmos. Meas. Tech., 12(4), 2219–2240, doi:10.5194/amt-12-2219-2019, 2019.

---

## Referee Comment (RC2) · Robert Healy (Referee) · 28 Aug 2019

Froyd and coauthors describe an approach to quantification of dust and specific aerosol species (organics, sulfate) using single particle mass spectrometer (SPMS) measurement data combined with concurrent scattering-based particle counting measurement data. The latter are used to provide accurate particle number-size distributions in bins as a starting point for the analysis. The approach involves first using mass spectral information to infer density and shape factor for each measured single particle. These properties are then used to convert the aerodynamic diameter of each particle to its respective volume equivalent diameter. Once all of the single particles are binned

into volume equivalent diameter bins, the fractional contributions of particle 'classes' in each bin are calculated. As the total particle number in each bin is already known as a function of time from the supporting particle counting measurements, here the SPMS data are only used to produce the breakdown of particle compositions within each size bin. Single particle and ensemble sulfate and organic mass concentrations can then be calculated. Total dust mass concentrations are also estimated. Although applied to aircraft datasets in this case, the approach could be extended to ground-based field studies for other SPMS instruments. The uncertainties associated with the method, including particle counting measurement uncertainty, SPMS counting statistics, the use of binned diameters, and assumptions around particle shape factor and density are carefully considered and laid out in detail. This manuscript represents a roadmap for future SPMS users that aim to use their single particle data in a more quantitative way. Although it is a little long, the content here is certainly useful for future applications of the method. I have only minor comments below.

Apart from the uncertainties listed in the Appendix, one issue is particles that are not efficiently ionized (or ionized at all) by the SPMS, because these classes will be absent in the analysis. Differences in ionization efficiencies for different particle classes, or absent classes, will affect the relative counts and fractional contributions of each class in each bin if it is assumed that all particle classes are detected with equal efficiency. Are there known particle mixing state impacts on relative ionization efficiencies for the PALMS instrument? If so these should be discussed and an estimation of the impact of this phenomenon on the quantification uncertainty would be useful.

Page 2, line 25: Nitrate, ammonium and potassium have also been previously quantified for particle classes and for single particles using similar approaches for groundbased measurements (Healy et al. 2013, 2014). Those applications also share the similarity with this work of taking concurrent particle counting measurements as the representative total number-size distribution rather than using size-dependent detection efficiency curves to work up from SPMS counts. It should be noted that that dataset was less challenging for quantification applications however, because only the submicron distribution was considered and crustal/sea salt contributions were minimal in that case. The assumption of equal detection efficiency for all mixing states was also taken in that work, but the spread in relative sensitivities observed for quantified species for each hour of the measurement period indicated that matrix effects associated with mixing state do impact quantification accuracy, at least for ATOFMS measurements.

Page 4, line 8: also 50% for 3.2  $\mu$ m?

Page 4, line 33: Are the negative spectra used in any way?

Page 15, line 15: Nitrate can be quantified using the approaches laid out here.

Page 16, line 8: "with decreasing altitude"

Page 17, line 19: Fig. 10 not included

References

Healy, R. M., Sciare, J., Poulain, L., Crippa, M., Wiedensohler, A., Prévôt, A. S. H., Baltensperger, U., Sarda-Estève, R., McGuire, M. L., Jeong, C.-H., McGillicuddy, E., O'Connor, I. P., Sodeau, J. R., Evans, G. J., and Wenger, J. C.: Quantitative determination of carbonaceous particle mixing state in Paris using single-particle mass spectrometer and aerosol mass spectrometer measurements, Atmos. Chem. Phys., 13, 9479-9496, https://doi.org/10.5194/acp-13-9479-2013, 2013.

Healy, R. M., Riemer, N., Wenger, J. C., Murphy, M., West, M., Poulain, L., Wiedensohler, A., O'Connor, I. P., McGillicuddy, E., Sodeau, J. R., and Evans, G. J.: Single particle diversity and mixing state measurements, Atmos. Chem. Phys., 14, 6289-6299, https://doi.org/10.5194/acp-14-6289-2014, 2014.

---

## Author Comment (AC1) · 3 Oct 2019

Please refer to the attached PDF for author responses.

Please also note the supplement to this comment:
https://www.atmos-meas-tech-discuss.net/amt-2019-165/amt-2019-165-AC1-supplement.pdf

---

## Author Comment (AC2) · 3 Oct 2019

Please refer to the Supplement for author responses.

Please also note the supplement to this comment:
https://www.atmos-meas-tech-discuss.net/amt-2019-165/amt-2019-165-AC2-supplement.pdf

———————————————————